# Functional regulation of aquaporin dynamics by lipid bilayer composition

Anh T. P. Nguyen [1], Austin T. Weigle [2] & Diwakar Shukla [1,3,4,5] ✉

With the diversity of lipid-protein interactions, any observed membrane protein dynamics or functions directly depend on the lipid bilayer selection. However, the implications of lipid bilayer choice are seldom considered unless characteristic lipid-protein interactions have been previously reported. Using molecular dynamics simulation, we characterize the effects of membrane embedding on plant aquaporin SoPIP2;1, which has no reported high-affinity lipid interactions. The regulatory impacts of a realistic lipid bilayer, and nine different homogeneous bilayers, on varying SoPIP2;1 dynamics are examined. We demonstrate that SoPIP2;1's structure, thermodynamics, kinetics, and water transport are altered as a function of each membrane construct's ensemble properties. Notably, the realistic bilayer provides stabilization of non-functional SoPIP2;1 metastable states. Hydrophobic mismatch and lipid order parameter calculations further explain how lipid ensemble properties manipulate SoPIP2;1 behavior. Our results illustrate the importance of careful bilayer selection when studying membrane proteins. To this end, we advise cautionary measures when performing membrane protein molecular dynamics simulations.

Transmembrane proteins are crucial vehicles for cellular transport and communication, composing ~20–30% of all proteomes[1] and ~60% of drug targets[2]. Understanding mechanisms governing the functional characteristics of these cellular machines can be leveraged for pharmaceutical[3] and agricultural applications[4]. Despite their sizable representation in the proteome, membrane proteins account for only ~1–2% of the total available structures in the Protein Data Bank (PDB)[5]. Even with advancements in cryo-EM and X-ray crystallography, membrane protein crystallization remains a highly technical task, including careful selection of efficient expression systems, suitable detergents for solubilization, and buffer conditions for purification and crystallization. These technicalities also hinder biophysical experiments for studying protein-lipid interactions, like direct structure resolution and FRET to characterize lipids bound to proteins, or NMR and FT-IR to study binding affinities[6]. State-of-the-art experimental instruments cannot resolve specific dynamics and atomistic interactions in spatiotemporal constraints without the help of computational methods.

Molecular dynamics (MD) has been incorporated into the membrane protein biophysics realm to provide a label-free, atomistic look at any membrane protein system. In particular, recent advances in enhanced sampling, high-performance computing, and methodological developments prove MD to be a highly applicable method for understanding membrane proteins[7].

Confidence and accuracy are critical for translating modeling and simulation results to experiments. Although biological membranes are highly complex and asymmetric with diverse lipid species, model bilayers are often assumed to be homogeneous and symmetric when used for in vitro experiments or in silico membrane models. These simple bilayers try to approximate ensemble averaged properties of cellular membranes (i.e., acyl chain unsaturation, lipid headgroup chemistry, sterol presence/absence). Whatever lipid species is deemed the most abundant for a given cell, or most necessary for a given protein, is selected and ultimately becomes a "default" choice for lipid reconstitution in (computational) experiments. For example, POPC is a

[1]Department of Chemical and Biomolecular Engineering, University of Illinois Urbana-Champaign, Urbana, IL 61801, USA. [2]Department of Chemistry, University of Illinois Urbana-Champaign, Urbana, IL 61801, USA. [3]Center for Biophysics and Quantitative Biology, University of Illinois Urbana-Champaign, Urbana, IL 61801, USA. [4]Department of Bioengineering, University of Illinois Urbana-Champaign, Urbana, IL 61801, USA. [5]Department of Plant Biology, University of Illinois Urbana-Champaign, Urbana, IL 61801, USA. ✉e-mail: diwakar.shukla@shuklagroup.org

popular lipid choice for mammalian cell membranes, while POPE/POPG is commonly used for bacterial membrane simulations[7]. For simulation studies, simplistic bilayer selections can be attributed to the lack of high-quality lipidomic data and computational difficulty.

However, membrane proteins' functions, cellular trafficking, and conformational dynamics are sensitive to the encompassing plasma membrane bilayer[8,9]. Recently, MD studies of lipid-protein interactions focus on proteins co-crystallized with lipids or those with known lipid binding sites (e.g., G-protein coupled receptors, solute carrier transporters, ion channels)[6]. The lipid bilayer's impact on membrane proteins can be through direct lipid-protein interactions (i.e., strong binding and specific molecular contacts) or non-direct modulation effects (i.e., global membrane properties such as curvature, thickness, and fluidity). For example, a suite of coarse-grained simulations on ten different proteins in a 6000-lipid membrane of more than 60 lipid species showed specific lipid enrichments local to the protein, generating a protein-specific lipid "fingerprint"[10]. As a result, choosing a membrane composition for modeling a membrane protein is important in interpreting simulation results because of the contributions of specifically recruited, as well as non-specifically bound, lipids.

There is a chance that in vitro experiments and MD simulations for membrane proteins may be performed with suboptimal bilayer conditions. Nevertheless, there has not been a systematic study on the universal consequences of embedding a single protein in a potentially non-native lipid bilayer. Gu and de Groot studied the effect of variable lipid tail unsaturation on MthK potassium channel dynamics, but only used PC headgroups[11]. Our recent work[12] and work from the Im group[13] characterized realistic bilayers for their respective species, but—given the complexity of realistic bilayers—even an "appropriate" lipid bilayer selection has the potential to introduce variable lipid conformations. When a greater amount of different lipid types is present in a bilayer, there is a greater probability that a membrane protein could be uniquely stabilized by different combinations and orientations of diverse surrounding lipids. These numerous lipid conformations can considerably impact reported membrane protein dynamics. Concern for how membrane composition choice affects conformational sampling and modeled functions of a protein without a priori known lipid binding interactions is even greater. Herein, we design a study to quantify the effects of selecting (non-native) lipid environments when simulating membrane proteins. Using a model membrane protein, aquaporin, we provide a systematic evaluation of different lipid bilayers' influence on one protein without specific lipid binding sites or interactions.

Aquaporin (AQP), a highly conserved transmembrane protein responsible for water transport and regulation, can be used as a model system for a few reasons. First, aquaporins are found in all kingdoms of life, from yeast and bacteria to eukaryotes[14,15]. Thus, regardless of sub-cellular localization, AQPs must be functional no matter the lipid environment. Due to this universality, AQPs do not depend on specific lipid interactions like cholesterol binding to G-protein coupled receptors (GPCRs)[16–18] or PIP2 to ion channels[19,20]. Second, AQP water transport function occurs on a shorter timescale than substrate transport and ligand binding seen in transporters or receptors. Therefore, characterizing water transport via unbiased MD is computationally tenable across many systems with different lipid bilayers. Third, AQPs are generally static channels that do not engage in large conformational changes for their function[15]. For example, transporters must go through the outward-facing, occluded, and inward-facing states for a full transport cycle. Plant AQPs, or specifically the structurally resolved spinach aquaporin (SoPIP2;1), have a notable conformational change in a cytosolic loop containing four to seven more residues than that of other AQPs (Supplementary Fig. 1)[21]. This structural hallmark of plant AQPs can open or close the water-transporting channel. Plants evolved this functionality for water regulation responses adapted against extreme drought or flooding conditions. Given its size, the conformational

change of the plant AQP loop is a process that can be reasonably captured within atomistic MD timescales. Few mechanistic modeling studies of non-photosynthetic plant membrane proteins have been completed[12,22–24]. Recently, a realistic plant membrane bilayer for molecular simulation has been characterized[12].

Long time-scale MD simulations of SoPIP2;1's opening and closing cycle can then be justified as a model system to understand the intrinsic effects of membrane choice on the computational study of a membrane protein. Herein, we report the influence of nine simplistic, homogeneous bilayers compared to a complex, heterogeneous membrane on the functions and opening/closing dynamics of model system SoPIP2;1 (Fig. 1). Through this study, we (1) offer quantitative and qualitative evaluations of SoPIP2;1 conformational dynamics and functions; (2) give higher spatial resolution to explain experimental observations of bilayer lipid composition on aquaporin; (3) provide precautions for membrane composition selection in studying membrane proteins; (4) demonstrate potential applications in leveraging lipid local environment for engineering desired membrane protein functions and conformational selections.

To summarize our key take-home points, we find that membrane choice induces different slowest processes, which will inherently alter thermodynamics, kinetics, and functional observations. However, more than one model bilayer can appropriately model target membrane protein function. We recommend that MD practitioners research known lipid-protein interactions to make the best system construction decisions. Ensemble average properties revealed from literature search or lipidomic data from a related organism/cell type should be represented in the modeled bilayer. The configuration of the annular shell(s) used to seed simulations should be diversified. MD practitioners should employ replicates with varied membrane packings to avoid artifacts caused by initialized configurations for both simple and complex bilayers. Realistic bilayers with asymmetric sterol distributions could cause tight bilayer packings that could drastically affect results. Lastly, we encourage researchers to confirm that computationally observed states are functional. Complementary analyses should be used to build trust in observed structures. When applying enhanced sampling techniques, caution must be exercised by inspecting starting states or using a combination of lipid bilayer and protein features to drive simulations.

## Results

### Aquaporin conformational dynamics

A protein sequence's conformational free energy landscape portrays the energy of every possible conformation projected onto the selected reaction coordinates. As most ensemble conformations are not stable enough to be structurally resolved under experimental conditions, atomistic MD simulation can reveal metastable states throughout global free energy landscapes to help understand the conformations of interest for a given process. From the simulation data, the constructed MSMs provide the stationary distribution of the sampled conformations for the free energy landscape. With mean first passage time, the kinetics for transition between the resolved open and closed crystal structures of SoPIP2;1 is elucidated given a membrane lipid bilayer (Figs. 1–3).

Figure 2 illustrates the MSM-weighted (stationary distribution probability applied onto each state) free energy landscape of the SoPIP2;1 opening/closing projected on the first two respective tICs for each SoPIP2:lipid system. Bootstrapped sampling errors of the free energy landscapes can be found in Supplementary Fig. 15. The residue-residue distance features most correlated to each respective tICs are indicated on the axes labels and located on the structure to the right of each landscape. As tIC dimensionality reduction combines features to maximize the autocorrelation time of the principal components, the distance features most correlated to the tICs are often the "slowest" process found in the trajectories. These slowest processes dominate

# a

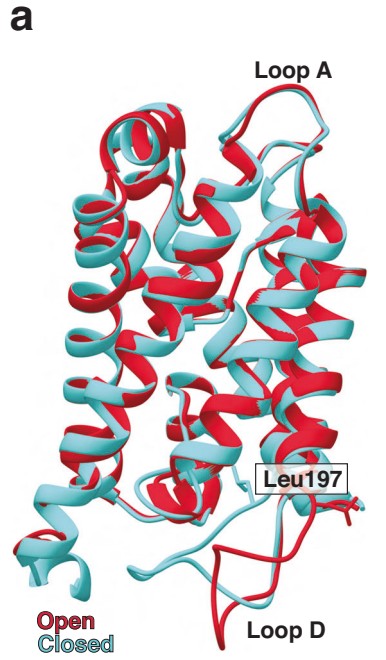

**Loop A**

**Leu197**

**Open**
**Closed**

**Loop D**

# b

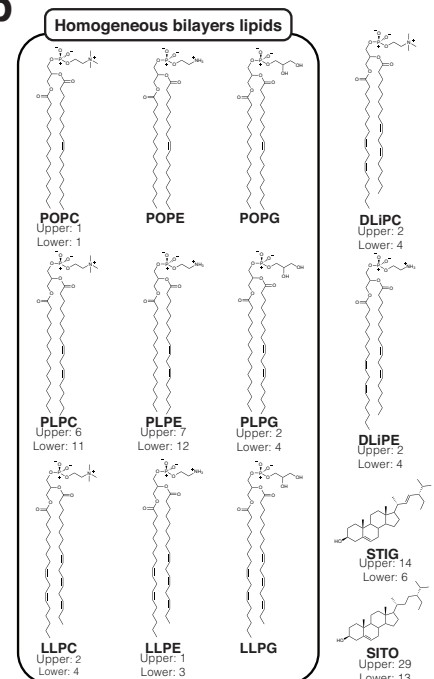

**Fig. 1 | System compositions of SoPIP2;1 molecular dynamics simulations.**
**a** Crystal structures of the open (PDB ID: 2B5F, red) and closed (PDB ID: 1Z98, blue) states of spinach aquaporin in the ribbon representation. Key differences between the structures are indicated, including the "plug residue" Leu197 (shown in the stick representation) and loop D. **b** Chemical structures of the lipids used and their composition in the complex lipid bilayer (if present). Enclosed in the box are the lipids used in the homogeneous bilayer systems, covering all three headgroups and varying levels of acyl chain unsaturation. POPC (1-palmitoyl-2-oleoyl-sn-glycero-3-phosphocholine; 16:0/18:1), POPE (1-palmitoyl-2-oleoyl-sn-glycero-3-phosphoethanolamine; 16:0/18:1), POPG (-palmitoyl-2-oleoyl-sn-glycero-3-phosphatidylglycerol; 16:0/18:1), PLPC (1-palmitoyl-2-linoleoyl-sn-glycero-3-phosphocholine; 16:0/18:2), PLPE (1-palmitoyl-2-linoleoyl-sn-glycero-3-phosphoethanolamine; 16:0/18:2), PLPG (1-palmitoyl-2-linoleoyl-sn-glycero-3-phosphatidylglycerol; 16:0/18:2), LLPC (1-linoleoyl-2-linolenoyl-sn-glycero-3-phosphocholine; 18:2/18:3), LLPE (1-linoleoyl-2-linolenoyl-sn-glycero-3-phosphoethanolamine; 18:2/18:3), LLPG (1-linoleoyl-2-linolenoyl-sn-glycero-3-phosphatidylglycerol; 18:2/18:3), DLiPC (1,2-dilinoleoyl-sn-glycero-3-phosphocholine; 18:2/18:2), DLiPE (1,2-dilinoleoyl-sn-glycero-3-phosphoethanolamine; 18:2/18:2), STIG (stigmasterol), and SITO (ß-sitosterol). For more information on lipid bilayer assembly and lipid names, please refer to our "Methods −System assembly" section.

the timescale of the simulation and are important for understanding protein conformational dynamics. With the usage of fully data-driven protein distance feature selection, MSMs produced different slowest processes for the same protein in each membrane bilayer system (gray structures in Fig. 2), suggesting membrane composition changed SoPIP2;1 dynamics. From Törnroth-Horsefield et al.[21], key gating mechanisms were hypothesized to involve loop D and hydrophobic pore plug Leu197 from structure data and nanosecond simulations. Here, microsecond timescale simulations inferred a diverse set of processes involved in the opening and closing of the SoPIP2;1 channel.

For nine out of ten systems, at least one of the two slowest processes involves residues of loop D and the hydrophobic pore, except for PLPE. Both slowest processes in the PLPE system involve Cys69 and/or Val68, located on the extracellular loop A connecting TM1 and TM2. Loop A is also involved in one of the slowest processes for SoPIP2;1 in the POPE, POPG, LLPE, LLPG, and complex systems. This could be due to loop A folding across the simulations, which is a process occurring in a timescale slower than loop D movement, depending on the lipid environment. However, even in the PLPE bilayer, the two slowest processes still correlate to the opening and closing of loop D, as the two crystal structures are separated across two opposite basins of the landscape. In general, the data-driven selection was still able to distinguish the open from the closed crystal structure in two separate minima (red and blue dots on the landscape of Fig. 2 for nine out of ten systems, except for the LLPE bilayer). While other bilayers prompt SoPIP2;1 to have the slowest processes correlated to the opening/closing transition, the LLPE bilayer selects a process completely different from the expected loop D movement. As

a result, the two crystal structures fall into the same minima of the LLPE free energy landscape.

The impact of lipid bilayer compositions on the structural equilibrium of SoPIP2;1 is evident from the varying placement of SoPIP2;1 crystal structures in or near the energetic minima. Protein structures determined in experiments are often embedded in non-native liposomes or detergents to be crystallized, and it has been shown that the obtained crystals of the same transporter are dependent on their lipid environment as well[25]. Thus, in different lipid bilayer membranes, crystal structures might not be the most stable conformations. SoPIP2;1 was crystallized in detergents containing polyethylene glycol[21]. Deviation in the relative stability of these crystal structure poses is apparent in the free energy landscape of SoPIP2;1 in different bilayers (Fig. 2). Both crystal structures are only stable (relative energy <1.0 ± 0.2 kcal mol[−1]) in the POPG and PLPC bilayers, and slightly stable (relative energy 1.0−2.0 kcal mol[−1]) in the POPC, PLPG, LLPC, and LLPE bilayers. Meanwhile, the complex bilayer, along with POPE, PLPE, and LLPG, depict at least one crystal structure state with a relative energy between 2.5−3.6 ± 0.2 kcal mol[−1]. Interestingly, the thermodynamics of SoPIP2;1 in the complex bilayer does not follow the same trend as our previous work[12]. In particular, the complex bilayer was able to stabilize intermediate conformational states for sugar transporter OsSWEET2b to facilitate faster transitions among states[12]. In the free energy landscape of SoPIP2;1 in the complex membrane, the barrier is high for transitioning among minima (-2.6−2.8 ± 0.2 kcal mol[−1]). A macrostate similar to the closed crystal structure is hyper-stabilized with a relative free energy smaller than 1.0 ± 0.2 kcal mol[−1] while all other macrostates have relative energy of 1.0−2.0 ± 0.2 kcal mol[−1]. In contrast, combinations of

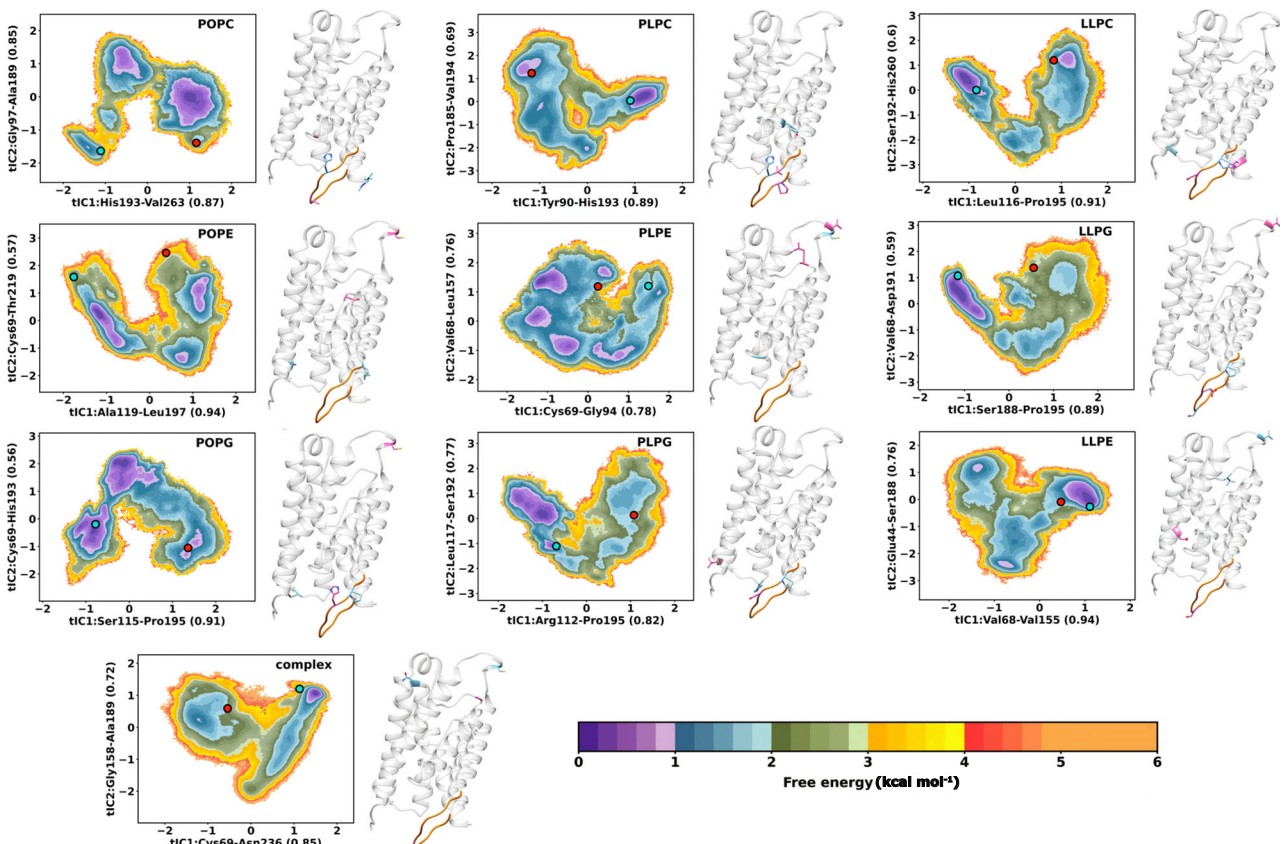

**Fig. 2 | Free energy of the SoPIP2;1 opening/closing transition from simulations.** MSM-weighted (stationary distribution applied) energy landscapes of SoPIP2;1 conformational changes in lipid bilayer systems (nine homogeneous and one complex membrane) projected onto the first two components of the time-lagged independent component analysis (tICA). The clusters most similar to the open and closed crystal structures are located on the landscapes as a red and blue dot, respectively. The distance feature most correlated to each component is indicated on the axes labels and located on the crystal structure to the right of each landscape. The residues involved in the first and second time-lagged independent components (tIC1 and tIC2) are shown in the blue and pink stick representations, respectively. Loop D is highlighted in orange as a reference. The colorbars represent free energy expressed as kcal mol$^{-1}$.

anionic headgroup PG with lower unsaturated tails (PL and PO); zwitterionic bulky headgroup PC with any acyl tails; and zwitterionic amine headgroup PE with high tail unsaturation (LL) can stabilize the SoPIP2;1 crystal structures as the near-lowest energy metastable states in MD simulations.

Despite being well discretized, the crystal structure poses shown in the tICA landscapes for POPE, PLPE, LLPG, and the complex bilayer are not always the most stable conformations (Fig. 2). Each of these SoPIP2:lipid embeddings share having loop A dynamics as one of the dominant tIC components. POPG and LLPE are also dominated by loop A dynamics, although POPG crystal structures are indeed the most stable. Conversely, LLPE tIC decomposition places both crystal structures within the same minima. For plasma membrane intrinsic protein (PIP) aquaporins, like SoPIP2;1, loop A participates in tetramer formation[26]. Loop A, and its homologous sequences in non-PIP orthodox AQPs, typically engage in direct hydrogen bonding contacts and/or disulfide bridges to help with monomer association and tetramer assembly[27]. AQP proteins are ubiquitously found as tetramers, but each monomer constitutes a functionally independent pore[28]. We used SoPIP2;1 monomers to complete this study due to the computational efficiency and literature-based justification (see "Methods−System assembly"). Given the dependence on loop A dynamics, it is likely that each of these systems would better stabilize the SoPIP2;1 crystal structures when modeled as a tetramer. However, the difference in tIC components reinforces how some bilayers can better stabilize SoPIP2;1 than others.

Figure 3 contains the calculated MFPT between the two crystal structure clusters on the free energy landscape. Error bars were computed from the 200 bootstrapped samples. Even though the zwitterionic bulky PC headgroup stabilized the crystal structures with relative free energy less than 2 kcal mol$^{-1}$, the conformational transition between the two structures in the POPC and LLPC bilayers proved difficult (transition of >200 μs). This could be due to the high energy of the transition regions, which are the highest among all SoPIP2:bilayer constructs with the peak relative energy of 3.2−3.4 ± 0.2 kcal mol$^{-1}$. In other bilayers, SoPIP2;1 has transitions between the open and closed conformations to occur within 100 μs, matching the timescale of loop movement[29]. *In planta*, the open conformations are hypothesized to be a more frequently adopted conformation, whereas the closed conformation should only be present during extreme abiotic conditions[15]. Although a closed-like state for SoPIP2;1 can be presumed as less favored under basal or equilibrium conditions, this preference cannot be differentiated as an intrinsic protein property or a consequence of various biochemical pathways within a plant cell. However, the MFPT calculations demonstrate a difference between the kinetics and the thermodynamic implications on the model protein due to membrane selection. Therefore, our main takeaway is that SoPIP2;1 in most bilayers exhibits opening/closing transitionary kinetics within a similar order of magnitude. Considering thermodynamics, nearly all SoPIP2:bilayer conformational landscapes, except for that of the complex bilayer, offer greater stabilization for the closed than the open state (Fig. 2). The transition rate of SoPIP2;1 in POPC, POPG, PLPC, PLPE, LLPC, and LLPE bilayers is slower for closing than opening. These same bilayers can also, at least relatively, stabilize the crystal structures in each of their respective free energy landscapes (Fig. 2). Meanwhile,

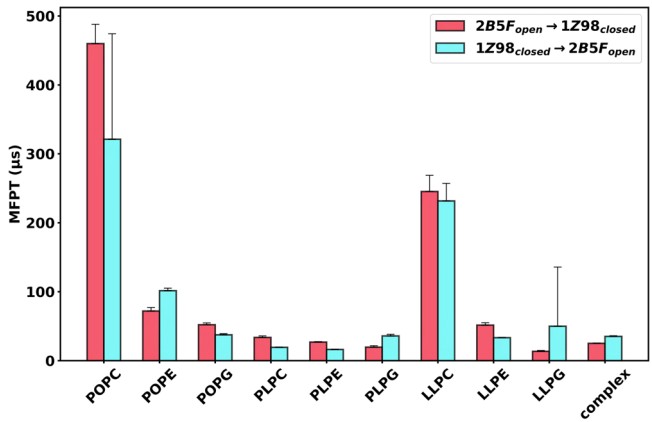

**Fig. 3 | Mean first passage time (MFPT) of SoPIP2;1 opening/closing transitions.** MFPT of the transition between the crystal structure clusters in the landscapes of Fig. 2. Data are presented as mean values ± SEM of 200 bootstrapped samples. Individual data distributions for each MFPT calculation are provided in Supplementary Fig. 16.

simulations of SoPIP2;1 embedded in the POPE, PLPG, LLPG, and complex bilayers show a faster closing MFPT. Each of these bilayers shifts at least one of the SoPIP2;1 closed and open crystal structures away from the energy minima. As a result, membrane selection imposes varying conformational samplings of SoPIP2;1, along with thermodynamic stability and transitionary kinetics for SoPIP2;1 crystal structures.

Differences in relative crystal structure stability among the SoPIP2:bilayer embeddings prompted the question of whether the conformational transition mechanism of SoPIP2;1 varies among membrane systems. Specifically, characterization of the open- versus closed-like character of SoPIP2;1 intermediate states can measure the thermodynamic control in-between gating poses by each bilayer. To do so, 1000 randomly selected frames in each lowest energy minimum (relative energy <1.5 kcal mol⁻¹) are extracted from the trajectories. Structures acquired from each frame were quantitatively assessed by measuring the dihedral angle of the pore plug Leu197 with respect to TM4/TM5 (Fig. 4). Because Leu197 is buried inside the pore for the closed conformation and extended outside into the membrane in the open conformation, Leu197 serves as a good indication of the opening or closing transition of the channel. The four atoms involved in the dihedral angle calculation are represented as balls on the open and closed crystal structures in Fig. 4a. A simplified visualization of the angle is illustrated in Fig. 4b. The distributions of this angle for each of the 1000-frame macrostates are presented as violin plots in Fig. 4c.

Impacts of the bilayer selection on the protein dynamics are observed clearly in the varying open- or closed-like character of Leu197 across intermediate states. In systems with one intermediate, this state can be induced by the membrane to be closed-like (POPG, complex), open-like (POPE), or somewhere in the middle (LLPE). Systems with multiple intermediate states adapt to both the open and closed states. Specifically, if the dihedral transition is described as a reaction coordinate of the transition, there can be a skew toward the open or closed state. In the PC and PG headgroups-containing bilayers, the reaction coordinate is skewed to the closed-like states. For the PE membranes, the intermediates favor having an open-like pore plug. Comparing the acyl tails, a clear observation lies in the LL-tail bilayers having intermediates sharing characteristics of both the open and closed Leu197 plug. The high unsaturation of the LL acyl tails provides the membrane with more flexibility and fluidity (discussed in more detail in the "Membrane properties" section), thus giving the protein a "smoother" transition pathway. Overall, the transition path from closed to open varies drastically for the same protein embedded in different membrane bilayers.

## Water transport function

With the structural and dynamical differences in SoPIP2;1 gating and pore plug orientation resulting from membrane embeddings, the water transport function was also examined. Three separate trajectories occupying each minimum of the free energy landscape were selected (see Supplementary Fig. 17 for the exact position along the respective tICA landscapes). An exception applies for the SoPIP2:LLPE landscape, where the open and closed crystal structures of SoPIP2;1 lie in the same minima. Thus, two sets of trajectories in this SoPIP2:LLPE minimum were selected, each set corresponding to a crystal structure. Adapted from Gelenter et al.[30] we calculated the number and rate of water transport events throughout each given 100-ns trajectory. Each transport event includes a water molecule traveling from the extracellular bulk, through the channel pore, then into the intracellular bulk. We report the number of waters imported because it directly correlates to the number of waters exported (Supplementary Fig. 19a), as aquaporins are bi-directional water channels. The water import and export rates reveal Poisson distributions, where increases in water transport events are greatly skewed toward open structures (Supplementary Fig. 19c, d). Thus, evaluating the number of transported waters serves as the best quantitative measurement for differentiating transport activity between SoPIP2:bilayer embeddings.

The number of waters imported by SoPIP2;1 for the open-like and closed-like states are reported in Fig. 5a. Regardless of whether the crystal structures are stabilized inside the lowest energy regions, water transport, or possibly water leakage, can still occur for the closed-like states in most of the bilayer systems. Overall, this suggests that although lipid bilayers can stabilize a structurally closed SoPIP2;1 conformation, these resulting states may not necessarily be able to function as closed aquaporins. To this end, only the POPG, LLPE, and complex bilayers were able to stabilize a functionally closed, non-transporting conformation of SoPIP2;1 within a structurally closed-like intermediate or fully closed macrostate minima. Conversely, other bilayers stabilize partially open or fully open conformations of SoPIP2;1. A reasonable expectation would be for the water conductivity to be higher in open-like states than in closed-like states, which is observed for only the POPG, PLPE, PLPG, and LLPE bilayers. The agreement makes sense for LLPE, as the selected trajectories directly come from the location of the crystal structures on the free energy landscape (Fig. 2). Additionally, the same trend where the combination of anionic headgroup PG and low unsaturation acyl tails match the expected crystal structure stabilization (Fig. 2, discussed in the previous section) is seen in the expected water transport activity. On the other hand, reversed from the free energy landscapes—where the bulky zwitterionic PC headgroup in combination with acyl tails of any unsaturation degree can stabilize the crystal structures as expected—the PC-containing lipid bilayers failed to preserve the expected relationship of water transport between open- and closed-like structures.

Akin to protein-ligand interaction studies, aquaporins must not only transport, but also have pore-lining residues bind to, water molecules. To verify the water-conducting ability of the SoPIP2;1 pore in response to lipid bilayer insertion, the number of waters occupying the pore at each frame per trajectory was computed, then averaged over the three trajectories of each respective macrostate (Fig. 5b). Overall, all SoPIP2;1 channels are conducting water into the pore, with 15–40 water molecules inside the channel at all times. Additionally, for lipid bilayer systems with SoPIP2;1 having expected transport activity (i.e., more water transport in the open-like states than the closed-like states), the number of waters occupying the pore of the open and closed states are stable and similar (~20 for POPG, ~15 for PLPE, ~12 for PLPG). Moreover, the intermediate states of these systems have a higher number of waters occupying the pore compared to the closed/open states (~25–30 for POPG, ~20–25 for PLPE, ~20–35 for PLPG). These intermediates also produce high fluctuation in the number of

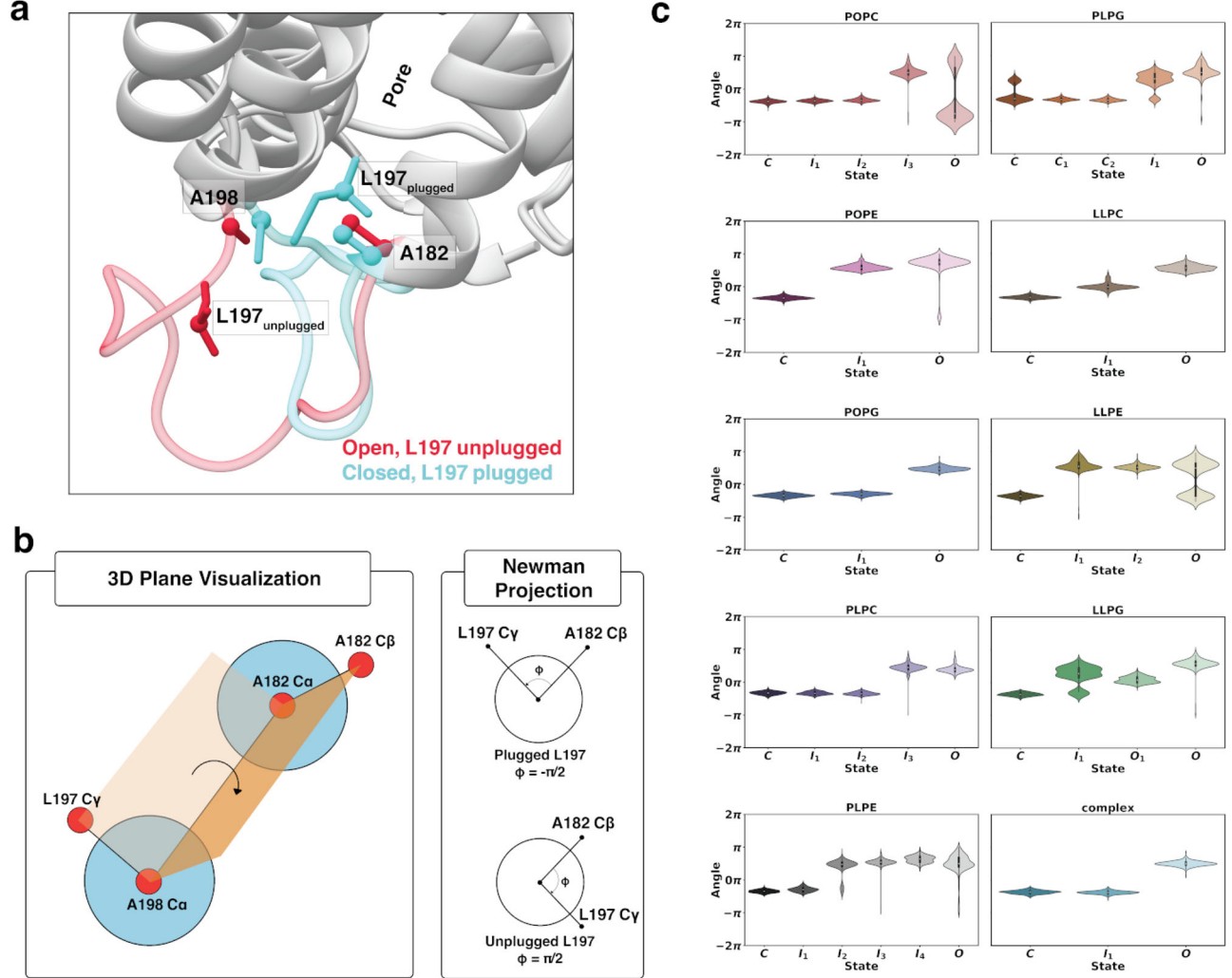

**Fig. 4 | Characterization of pore plug Leu197 in each of the respective SoPIP2:bilayer macrostates. a** Atoms used in the dihedral angle calculation shown on the crystal structures of the open (PDB ID: 2B5F, pink) and closed (PDB ID: 1Z98, blue) states. Residues are shown in the stick representation, and atoms involved in the calculations are shown in the ball representation. **b** Simplified schematics of the dihedral angle between the $C_\gamma$ of Leu197 and $C_\beta$ of Ala182. **c** Violin plots of the dihedral angle in each macrostate of each lipid bilayer system. Distributions in (**c**) are calculated using 1000 randomly selected samples (frames) from each respective metastable state energy minima. Data in (**c**) are presented as mean values ± SEM.

waters inside the channel (±5 for POPG, ±10 for PLPE, ±15 for PLPG). All PC-containing systems produce high fluctuation in every macrostate of SoPIP2;1 (from ±5 for POPC and LLPC to ±15 for PLPC). Uniquely, in the complex bilayer, all three states are stable and have around the same number of waters occupying the pore through the trajectories (from 15–25 molecules). This aligns with the stabilizing effect of the complex bilayer for OsSWEET2b shown in our previous work[12]; however, for SoPIP2;1, this bilayer is stabilizing a non-functioning state.

Overall, the computed number of water molecules transported is within three orders of magnitude from the experimental literature value of $10^4$ waters per 100 ns for SoPIP2;1 in the *E. coli* liposome[31] or 100–200 waters per 100 ns for aquaporin in general[32,33]. Most strikingly, the complex membrane hinders all water transport across all SoPIP2;1 macrostates with virtually no water transported in both the open-like and closed-like states. Our functional water transport analyses demonstrate that the conformation of loop D does not necessarily determine the water conductivity of the channel or overall transport activity. Because all SoPIP2;1 macrostates have 15–40 water molecules occupying the pore at all times, we suspected strong interactions inside the pore preventing water from entering the cytosolic region. The interactions are possibly a response to some

protein-lipid interactions, thereby preventing water from passing into the intracellular side to complete its transport cycle.

Across selected macrostate trajectories, the SoPIP2;1 channel is conducting water, so a blockage must be preventing water from going through the pore region and into the intracellular bulk. We selected two trajectories in the open loop D conformation with the two extreme transport activities (117 waters imported in LLPE, orange, versus no water imported in complex, blue) to characterize the channel structure in Fig. 6. In AQP, there are two conserved regions inside the pore which govern the orientation of the water molecules to prevent proton transfer: the ar/R (arginine/aromatic) selectivity filter and the NPA (Asn, Pro, Ala) motif. The HOLE radius calculations provide the approximate pore size at a given relative z position, so we contextualize the z positions by finding the $C_\beta$ positions of key pore-occluding residues (the ar/R selectivity filter, NPA motif, and coil occluding the pore) along the *z*-axis (Fig. 6a). As expected, in the extracellular side of the protein (positive z), the HOLE radius of the pore is the same for both transport cases. However, nearby Thr219 (of the ar/R selectivity filter) occupies completely different z-positions in the channel for each transport case (Fig. 6a, bottom panel).

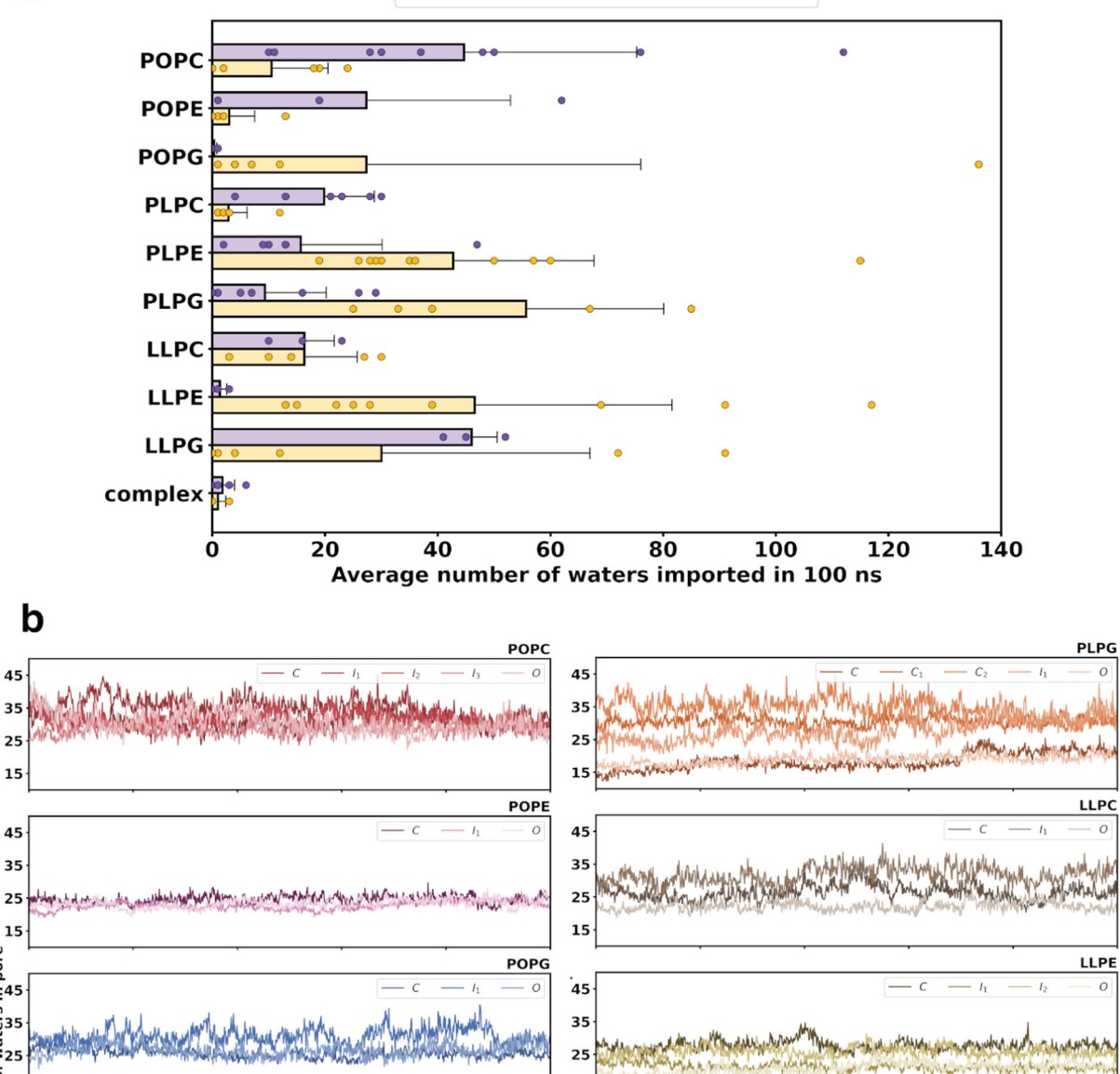

**Fig. 5 | Water transport activity of SoPIP2;1. a** Average number of waters transported per 100-ns trajectory for the open-like (yellow) and closed-like (purple) SoPIP2;1 macrostates. Errors are calculated as standard deviations among the closed-like or open-like trajectories. **b** Time evolution of the average number of waters occupying the protein pore of each macrostate in each bilayer system. Water transport data shown here was calculated using $n = 3$, where three independent continuous 100-ns trajectories (10,000 frames) were selected from each metastable free energy state per SoPIP2:lipid system. For reported water transport values in (**a**), data were grouped based off the relative open-like or closed-like character of the metastable state trajectories. Data in (**a**) are presented as mean values ± SEM.

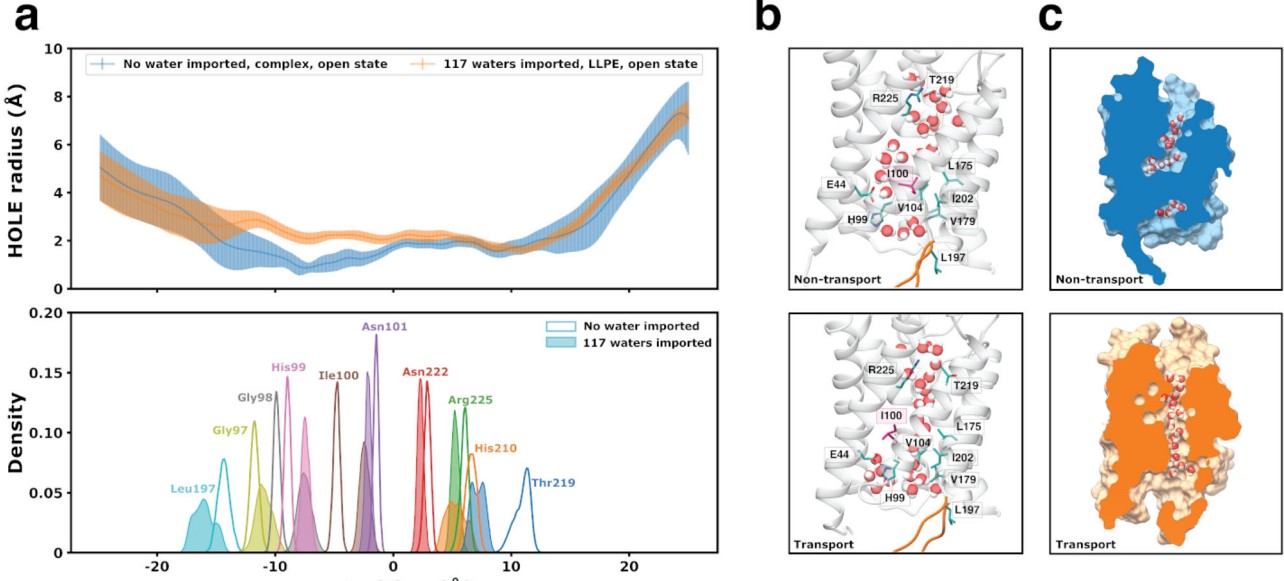

**Fig. 6 | Structural comparison of two extreme transport cases for the SoPIP2;1 open state embedded in two different lipid bilayers system (LLPE and complex). a** Pore HOLE radius along the z-position of the SoPIP2;1 protein pore aligned with the distribution of the $C_\beta$ z-positions in key residues inside the protein pore (the ar/R selectivity filter, NPA motif, and coil occluding the pore). The error bar for the radius at each z position is calculated as the standard deviation of the 10,000 frames of the trajectory. **b** Hydrogen bonding and hydrophobic interaction of key residues most different between the non-transport (top) and transport (bottom) cases. Highlighted residues are shown in stick representations. Leu197 and loop D are also shown as a reference point. **c** Cross section of the SoPIP2;1 pore surface indicating water blockage of the non-transporting open SoPIP2;1 (top, blue) and transporting open SoPIP2;1 (bottom, orange). Data in (**a**) are presented as mean values across the trajectory for each given point in the pore ± SD.

Moving down to the intracellular side, the most drastic difference in the HOLE radius of the two transport cases appears in the −12–0 z positions. In this region, the transport pore is ~1 Å larger in radius compared to the non-transport pore. Moreover, the radius of this region in the non-transport channel falls below 1 Å, which is smaller than ~1.7 Å, the van der Waals radius of a water molecule[34]. Thus, this region of the non-transporting pore is inaccessible to water. Structurally aligning this region with the key residues, we observe the difference in Ile100, His99, and Gly98 between the transporting SoPIP2;1 and non-transporting SoPIP2;1.

Therefore, we visualized interactions formed by Thr219, Ile100, and His99 (Fig. 6b). In the ar/R selectivity region, Thr219 is forming hydrogen bonds with Arg225 and pointing upwards in the non-transport pore, resulting in water entry as bulk solution. However, Thr219 breaks this hydrogen bond in the transport pore and points downwards, allowing water to proceed deeper into the channel in a single, ordered line. Nonetheless, this is not a requirement for a functioning SoPIP2;1. In Supplementary Fig. 20, the closed SoPIP2;1 structure in the POPC membrane has 62 water molecules transported in 100 ns, and the water molecules at the top of the pore are not ordered. It can be concluded that if the water molecules enter in an ordered manner, the pore is a functioning channel.

The key residue regulating transport is observed to be Ile100 (colored pink in Fig. 6b), belonging to a coil connecting TM2 and TMB, which occludes the pore. Ile100 forms a network of hydrophobic interactions that prevent the water from transporting through the channel into the intracellular region. This observation matched with other trajectories found to be non-transporting (Supplementary Fig. 20; open SoPIP2;1 in POPC, open SoPIP2;1 in PLPC). When Ile100 extends into the transport pathway, the linear water column is broken. As a result, the water attempts to find an alternative pathway for conduction by avoiding the hydrophobic region and redirecting toward the TM1/TM3/TMB vestibule. However, the hydrogen bonding between His99 and Glu44 (and possibly other interactions) prevents water from exiting the pore through that alternative pathway. It appears that certain lipid environment embeddings cause loop D opening and closure to no longer be a deciding factor for whether SoPIP2;1 can transport water.

Additionally, aquaporin and SoPIP2;1 literature described the water-conducting pathway as a single-file line, straight column during nanosecond simulations[21]. Here, we uncover another, although largely unsuccessful, transport pathway adapted by water molecules in a stabilized conformation of the SoPIP2;1 that is functionally depleting through long-timescale microsecond simulations. This hypothesis matches the water residence time analysis presented in Supplementary Fig. 21, which captures the average natural log of the residence time that each water molecule continuously spends in a given region of the pore. The pore regions were defined by laterally splitting the water-conducting channel into eight slices, each corresponding to a cylinder of width 8 Å, height 4 Å, and center defined by the center of geometry of 3–4 residues on the transmembrane helices. The residues and respective slices can be found in the representative structure in Supplementary Fig. 21. Specifically, in the open and closed state of SoPIP2;1 in the complex bilayer, the time that water molecules spend at Slices 4–6, which is the region of hydrophobic blockage, is similar to other bilayers ($-e^{4.2}$ or ~67 ps). There is a spike in residence time at Slice 6 of the intermediate state, possibly due to a water molecule stuck in the opening of the alternative pathway. Though in general, the remaining trends in water residence time are similar across each SoPIP2:bilayer system. But whether in a transporting or non-transporting SoPIP2;1 state, the waters present at each Slice are experiencing constant positional flux. This high fluctuation and constant movement in pore-residing water molecules showcase attempts at finding an optimal pathway of transport regardless of current channel conformation.

## Lipid binding interactions

Whether intracellular rearrangements seen in non-functional SoPIP2;1 states were caused by direct protein-lipid interactions was examined. With the pioneering advances in membrane protein simulation of the 1990s and the beginning of AQP structure determination

in the early 2000s, there is a rich computational history on AQP-lipid interactions[35]. Foundational simulation studies examining the AQP-lipid interface have found protein-lipid interactions to be consistent with crystal and electron densities, showing conservation of specific protein-lipid interaction sites without high-specificity binding[36–39]. Lateral exchange of annular shell lipids with the bulk has therefore been suggested as a regulatory mechanism for AQP function[39]. Conserved channel residue rearrangements have also been posited as a regulatory means to affect AQP water transport[28,39]. Perhaps lipid binding itself could be the fundamental driver of either proposed regulatory mechanism.

Comparing lipid binding interactions from our 100-ns metastable state trajectories, SoPIP2;1 was found to have an identical lipid binding interface compared to 52 previously simulated AQP proteins (Supplementary Figs. 22–24)[40]. Likewise, high-specificity lipid binding sites are also presumed to be absent in SoPIP2;1. Average lipid residence times between SoPIP2;1 open, closed, and intermediate states across the lipid bilayers are nearly identical as well (Supplementary Figs. 25–27). Out of the residues involved in the inhibitory channel rearrangements, only Leu197 and Ile202 have consistent interactions with lipids. Meanwhile, Leu175 has moderate lipid interactions except for POPC, POPG and LLPC simulations.

Given that Leu197 demonstrated strong interactions with each lipid bilayer, we generated state-specific loop D-lipid interaction fingerprints (Supplementary Fig. 28). Nearly all bilayers maintain strong interactions with loop D residues Leu197 and Ala198, showing average residence times greater than 70 ns. Loop D maintains the most lipid interactions when surrounded by POPE, followed by PLPE. Otherwise, most bilayer compositions only strongly interact with ~5 out of the 16 loop D residues. While SoPIP2;1 is in the closed state, loop D lipid interactions are minimal, as the loop is occluding the pore. Overall, our resulting fingerprints demonstrate diverse binding signatures arise when using different bilayers.

## Membrane properties

Despite using the same protein throughout all simulations, each lipid bilayer exhibits altered protein conformational changes, structural dynamics of transition, and stabilization of functional/non-functional states. What biophysical properties of the membrane bilayer may be responsible for inducing the changes seen in the SoPIP2;1 channel and water transport activity? Hydrophobic mismatch is defined as the difference between the membrane bilayer thickness and the hydrophobic thickness of the transmembrane protein. Naturally, membrane lipids' acyl chains have to adapt to the given hydrophobic length of the protein, the protein adapts to the bilayer, or both, to reduce the mismatch and minimize solvent exposure for the hydrophobic region of the protein[41–43]. Additionally, increasing interest in synthetic proteins alludes to the choice of liposomes or lipid bilayers in which to embed the proteins[44]. Understanding whether a membrane bilayer may adapt to a membrane protein can be leveraged to deplete or induce wanted behaviors or dynamics of the target membrane protein[45]. Moreover, the ability of a membrane bilayer to adapt to the hydrophobic thickness of the protein is often observed in the lipids directly surrounding the protein, which is known as the annular shell. Here, we quantify the ability of the annular shell lipids to change from the bulk membrane with respect to the protein by comparing the protein-bulk thickness difference and the annular shell-bulk thickness difference (Fig. 7). The thickness values calculated for each frame are illustrated in Fig. 7a, where the center rectangle represents the hydrophobic region of the protein and the beads with two tails represent the lipids. The lipids in the bulk section are colored orange, and the lipids in the annular shell are colored brown. The thickness calculation procedure is discussed in "Methods".

We compared the distributions of the thickness difference between the protein-bulk lipids (light green violin plots) and the difference

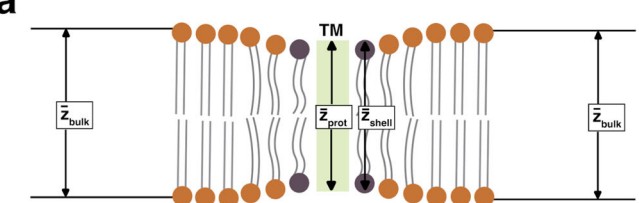

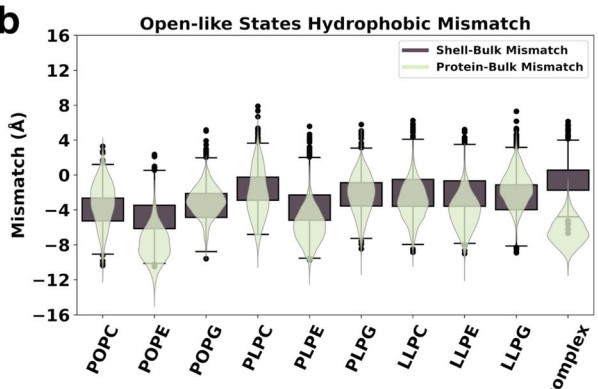

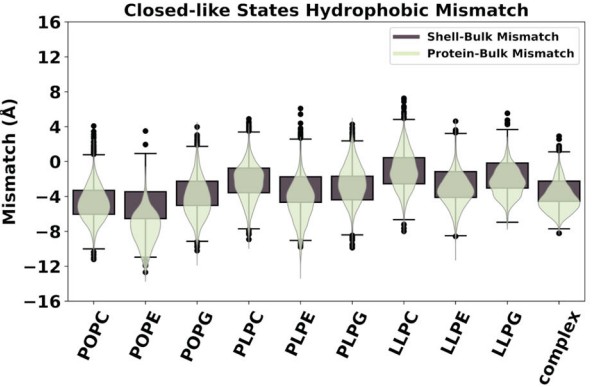

**Fig. 7 | Mismatch in thickness of SoPIP2;1 and the bilayer. a** Schematics of protein thickness, annular shell thickness, and bulk membrane thickness. Mismatch is calculated as the difference between the protein thickness and the bulk membrane thickness, along with the difference between the annular shell thickness and the bulk membrane thickness. **b** Violin plots of the protein-bulk mismatch (light green) and box plots of the shell-bulk mismatch (dark brown) for the open- and closed-like macrostates of each bilayer system. Distributions in (**b**) are calculated using 1000 randomly selected samples (frames) from each respective metastable state energy minima. Data in (**b**) are presented as mean values ± SEM.

between the annular shell-bulk lipids (dark brown box plots) for the 1000 frames randomly sampled in each SoPIP2:bilayer macrostate. The open-like and closed-like hydrophobic mismatch distributions are shown in Fig. 7b. Three scenarios can be observed. (1) If the distribution in the violin plots matches the distribution in the box plots, the annular shell lipids are able to adapt their conformation to cover the hydrophobic surface of SoPIP2;1. This perfect matching is the ideal condition for the membrane insertion of proteins. (2) If the violin distribution is lower than the box distribution, the annular shell lipids are less amenable to the protein. Still, the whole membrane can compensate for the protein's hydrophobic length through ensemble-induced curvature or deformation. (3) If the violin distribution is above the box distribution, neither lipids in the annular shell nor the bulk membrane are suitable for the protein due to their inability to accommodate the hydrophobic length of the protein. As a result, the hydrophobic surface of the protein is exposed to the solvent, which can be detrimental to its function and dynamics.

Overall, Fig. 7b demonstrates that the protein-bulk difference matches or is lower than the shell-bulk difference (scenarios (1) and (2)), suggesting that the lipids used in this study can adapt to SoPIP2;1. This observation validates that our previous characterization of the plant complex membrane[12] should be able to accommodate plant membrane proteins. The most striking difference among the systems can be seen in the complex bilayer of the open-like states, where the protein-bulk mismatch is much lower than the shell-bulk mismatch. The misalignment infers that the lipids occupying the annular shell cannot change their conformation from the bulk to match the protein in its open-like states. However, this complex membrane can still cover the protein's hydrophobic region. This mismatch could explain the limited water transport abilities of the open SoPIP2;1 in Fig. 5. Specifically, the packing of the complex membrane's annular shell in the open macrostate could have stabilized the hydrophobic blockage observed in Fig. 6, preventing the sampling of a functional channel. The addition of sterols (SITO and STIG) in large quantities (over 48% in composition) stiffens the membrane[46] due to aggregation into domains[47,48] and prevents the acyl tails of the phospholipids from changing its conformation or mixing in the membrane bulk. As such, the sampled conformation of the membrane traps its preferred conformation of the protein, leading to the highly stabilized non-functional open SoPIP2;1 state occupying the lowest energy basin. These results present a double-edged sword scenario where complex bilayer composition can essentially trap a protein within an unproductive conformation because of stabilization effects.

For the homogeneous bilayers, the violin and box distributions are either matching well or the violin distribution is slightly lower (in POPE, PLPE, and LLPE). When comparing the bilayers of the same acyl tails, the system containing the PE headgroups will always produce the most negative mismatch in shell-bulk and protein-bulk thickness. This observation correlates to PE-containing lipids' higher acyl tail order parameters (especially POPE) compared to lipids with PG or PC headgroups (Supplementary Fig. 29a). The PE headgroup is zwitterionic like the PC headgroup but contains an exposed quaternary amine, which can repel one another when packed tightly in a membrane, limiting the conformational changes that the tail can adapt to. Combining this effect with the least unsaturated tail, PO, the membrane becomes even less fluid. Thus, packing aquaporin in POPE can decrease its fluidity and stabilize a non-functional open state, matching its low water transport of open-like states in Fig. 5a. As the closed conformation can be adopted only during abiotic stress *in planta*, this intermediate state can be induced to allow the plants to conserve water even in the open conformation. Comparing the effect of the acyl tails for each headgroup, the highest unsaturation tail LL almost always has the least mismatch (except for PLPC and LLPC in the open-like states, where PLPC has less negative mismatch than LLPC). This observation also matches the comparison of lipid order parameters among unsaturation degrees in Supplementary Fig. 29b. As the higher unsaturation tail allows for spaces in the membrane packing, the bilayer becomes more flexible, and in general, enables the protein to move more freely. This additional fluidity induces more total water transport.

As the stabilization of non-functional open SoPIP2;1 in the complex bilayer was examined through hydrophobic mismatch, the occurrence of this state in some homogeneous bilayers requires further explanation. We analyzed the order parameter of each phospholipid spanning the membrane through each of the three continuous trajectories sampled within each SoPIP2;1 macrostate in its respective homogeneous bilayer (the selected trajectories are visualized in Supplementary Fig. 17). Earlier AQP simulation literature proposed bilayer thickness and ordering within the annular shell as a mechanism for regulating water transport[49]. In experiments, lipid order parameters can be calculated with quadrupolar splitting in $^1$H-NMR[50,51] or dipolar splitting $^{13}$C-NMR[52,53]. Lipid order parameters shed light on the general

order of the whole lipid membrane or the conformations adopted by each lipid molecule in the bilayer[54]. Here, we report the average order parameter of both tails in each lipid through each 100-ns macrostate trajectory with the coarse-grain order parameter $S_{cc}$ method in LiPyphillic[55]. The order parameter is calculated with Eq. (1), where $\theta$ is the angle between the membrane normal and the bonds connecting all two consecutive carbon atom pairs along the acyl tail. The brackets indicate an average over all carbon atoms in a given acyl chain:

$$S_{cc} = \frac{\langle 3cos^2\theta - 1 \rangle}{2} \quad (1)$$

Per Eq. (1), the order parameter of the phospholipid acyl chains ranges from −0.5 to 1, with −0.5 being the most disordered and 1 being the most ordered (completely linear tail, where $\theta = 180°$). Lower order parameter values reflect a fluid and flexible membrane; conversely, higher order parameter values suggest a stiff and rigid membrane. The average order parameter of all lipids in each trajectory was compared to the number of waters transported (Supplementary Fig. 30). However, the relationship between the average lipid order parameter and the number of waters transported is complicated. Instead, we seek to implement order parameter analyses to understand why some of the open states are non-functional. We omitted the closed states in this analysis because differences in transport activity amongst the closed states have otherwise been previously explained, as the membranes are unable to stabilize the fully closed crystal structures in the lowest energy minima. For each selected trajectory from the homogeneous bilayers, the position of each lipid molecule is projected onto the xy plane as a scatter plot colored by the average order parameter of the two acyl chains throughout the trajectory. In the background of the scatter plot is the heatmap of the protein's positions through the trajectory, colored by each of the transmembrane helices. Views of the helices on the structure can be found in Supplementary Fig. 1.

Figure 8 portrays twelve representative trajectories that vary in annular shell lipid order parameters and water transport activity. The selection of the trajectories is from scatter dots shown in Supplementary Fig. 31. At low transport activity (<30 waters imported), the shell lipids can adopt any order parameters from 0.14–0.25, which indicates a wide range of membrane rigidity. On the other hand, when the channel can transport more than 30 water molecules in 100 ns, the average order parameter of the shell negatively correlates to the number of waters transported (i.e., stiffer membranes transport less water). Most importantly, we note that no highly functional open SoPIP2;1 channel (>20 waters imported) can be found in bilayers of average order parameters higher than 0.19, corresponding to more rigid membrane constructs (Supplementary Fig. 31). Moreover, the highest and second highest shell lipid order parameters (most rigid bilayers) trajectories belong to SoPIP2;1 in the POPE and POPC bilayers. This observation is reasonable, as the PO acyl tail is monounsaturated, limiting the lipid's conformational flexibility. This does not apply to POPG, as one outlier SoPIP2:POPG trajectory transports over 130 water molecules (Supplementary Fig. 31). Thus, the zwitterionic phospholipid headgroups (PC, PE) combined with monounsaturated tails cause the stabilization of non-functional open states. If the anionic headgroup PG is combined with the PO tails, we observed a semi-stabilization of non-functional open states (SoPIP2;1 in one of the open state trajectories was able to reverse the hydrophobic block).

Despite having the same type of lipid species between the annular shell and bulk, the specific conformations of individual lipid molecules can have a significant effect on regulating protein function. Variation in relative lipid fluidity within a homogeneous bilayer can functionally mimic the introduction of a different lipid species altogether. This means that overall lipid bilayer configuration is critical to enabling, or even disabling, protein function. For all selected trajectories in the open states of Fig. 8, the lipids surrounding the protein have a

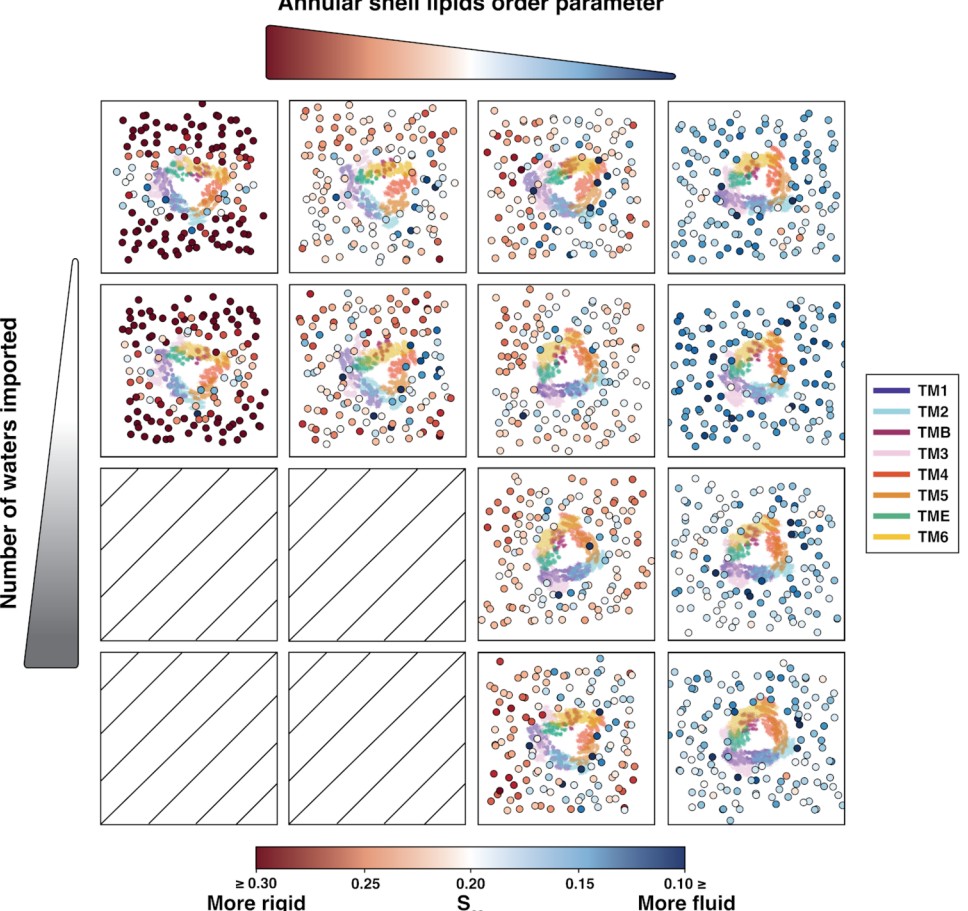

**Annular shell lipids order parameter**

**Fig. 8 | Water transport activity and lipid order parameter in the open states of the homogeneous bilayer systems.** Each square shows the projection on the xy plane of the lipids (scatter dots colored by average lipid order parameter, $S_{cc}$, bottom legend) and protein (heatmap colored by transmembrane helices, right legend) for one 100-ns representative trajectory of a given water conductivity and average annular shell order parameter. Empty squares indicate no trajectory found for high water transport activity and high lipid order parameters. The red-to-blue colorbar represents the $S_{cc}$ order parameter.

relatively lower order parameter than that of the lipids in the bulk, indicating all the selected lipids' abilities to adapt to SoPIP2;1. Comparing the position of the protein helices, the no-transport trajectories (top row) show how helical repositioning in response to the membrane environment results in pore contraction, which could be the reason that the hydrophobic block of Ile100 can become stabilized.

We also observed a positive linear correlation between the lipid order parameter and the average thickness for the homogeneous bilayer (Supplementary Fig. 32) across all selected macrostate trajectories. The LL-tail-containing bilayers' order parameter and thickness form one linear correlation, and the remaining bilayer forms another. As the lipid order parameter cannot be calculated for the complex membrane with sterols, we visualized the thickness as a function of the number of waters imported (Supplementary Fig. 33). Specifically, all SoPIP2;1 macrostates in the complex membrane have under ten waters transported. The thickness of the complex membrane lies between 40.5 and 41.5 Å, among the low-order POs and high-order PLs membranes. However, the correlation of the order parameter to thickness cannot be transferred from the homogeneous bilayers to a complex one. The hydrophobic mismatch demonstrated the packing near the open state to be detrimental to its function, and literature has shown an enrichment of sterols in the annular shell near aquaporin compared to the bulk membrane[10]. Additionally, experimental studies have shown that the addition of sterols reduces water transport in AQP0[56] and AQP4[57]. As a result, our observation for SoPIP2;1 functional depletion in the complex membrane matches experiments, opening potential implications

for the careful selection of model membrane bilayers in molecular dynamics simulations and benchtop experiments.

Moreover, we also observe that all SoPIP2;1 macrostates in the LLPC and PLPC bilayers transport under 30 water molecules per 100 ns (Supplementary Fig. 30), despite being able to stabilize both crystal structures in their conformational energy landscape (Fig. 2). Supplementary Fig. 29 separates the impacts of headgroups and acyl tails on water transport, membrane order, and membrane thickness. Overall, given the same acyl tail, the PE headgroup increases the tail order parameter and the membrane thickness, while the PO- and PC-containing lipids have similar order parameters. As expected, the LL tail always has the lowest order parameters, as it contains the greatest degrees of unsaturation. Except for PLPC, LLPC, and POPE, SoPIP2;1 macrostates in any other homogeneous bilayers can have varying levels of water transport, depending on the stabilization of the hydrophobic blockage inside the pore.

Lipid order parameter analysis illustrates that tuning of some membrane properties and lipid selection can control the activity of a given membrane protein. Specifically, for SoPIP2;1, embedding in membrane bilayers containing PC will induce low transport in the open macrostate despite adequately stabilizing the crystal structure conformations. Especially for PLPC and PLPC, all macrostates will experience lower transport activity (<30 molecules per 100 ns) than on average (~50 molecules per 100 ns). The PE headgroup increases the membrane's stiffness, thickness, and hydrophobic mismatch. When combined with a monounsaturated tail PO, the PE headgroup

significantly reduces water transport to under 20 molecules per 100 ns (one outlier at 60 molecules per 100 ns). Our calculations of reduced water transport agree with our loop D fingerprint analysis, as the stiffened POPE bilayer sustains many long-lasting loop D interactions (Supplementary Fig. 28). A charged PG headgroup combined with monounsaturated tail PO also leads to mostly low transport in the open macrostate (under 10 molecules per 100 ns, with one outlier at 139 molecules per 100 ns). The complex bilayer, with a high sterol composition of 48%, completely impedes water transport in all SoPIP2;1 macrostates. A sterol composition ≥40 mol% is required to maintain bilayer integrity in realistic cell membranes[58]. Therefore, for SoPIP2;1, to achieve a stable, expected behavior of transport and conformational dynamics, one should disrupt sterol crystallization by diversifying the initial bilayer configuration or use homogeneous bilayers containing lipids with polyunsaturated tails (PL or LL) and charged headgroup (PG) or zwitterionic, non-bulky headgroup (PE).

## Discussion

We demonstrated the various regulatory impacts of lipid bilayer selections on the structural dynamics, kinetics transitions, thermodynamics stabilization, and functional water conduction of spinach aquaporin, SoPIP2;1. By analyzing SoPIP2;1 dynamics and functions when embedded in nine homogeneous lipid bilayer membranes compared to one complex, heterogeneous plant membrane, we can uncover certain precautions for researchers when selecting an appropriate membrane lipid composition for computationally modeling or experimentally evaluating a membrane protein of interest. A summary of these points was provided at the end of the Introduction. We now go into further detail below.

In particular, each bilayer induces different slowest processes for SoPIP2;1 across simulation trajectories, indicating that the surrounding lipid environment does indeed affect conformational sampling of SoPIP2;1 to variable extents. To this end, effective SoPIP2;1 crystal structure stabilization similarly relies on membrane choice. Particularly, combinations of anionic headgroup PG with lower unsaturated tails; zwitterionic bulky headgroup PC with any acyl tails; and zwitterionic amine headgroup PE with high tail unsaturation can stabilize the SoPIP2;1 crystal structure as the near-lowest energy metastable states (relative energy <2.0 ± 0.2 kcal mol$^{-1}$). All of these bilayers prompt SoPIP2;1 to have a slower closing transition kinetics than for opening. For the complex bilayer, a macrostate similar to the closed crystal structure has the lowest relative energy of less than 1.0 ± 0.2 kcal mol$^{-1}$, while other macrostates are from 1.0–2.0 ± 0.2 kcal mol$^{-1}$. In the complex bilayer, along with POPE, PLPG, and LLPG, SoPIP2;1 presents a faster closing transition. Structurally, membrane selection influences the transitory pathways of SoPIP2;1 in the opening/closing processes. The open- or closed-like character of "pore plug" Leu197 among intermediate macrostates varies based on the lipid bilayer, suggesting that the lipid environment can alter reaction coordinate progress through selective structural stabilization. In the PC and PG headgroups-containing bilayers, the SoPIP2;1 intermediate states are skewed to have more closed-like states. Conversely, in the PE-containing bilayers, SoPIP2;1 intermediates favor a more open-like plug. Bilayers with the LL acyl tail are able to sample both the open- and closed-like characters equally, likely due to LL acyl tail flexibility caused by a high degree of unsaturation.

While SoPIP2;1 conducts water inside the pore and those water molecules are constantly traversing across pore regions in all bilayer embeddings, certain bilayers impede water transport of all macrostates (complex membrane), limit transport in the open-like states (POPE, PLPC), or produce leaky closed-like states (all except for POPG and LLPE). We uncovered how the loop D conformation does not directly alter the water conductivity of the SoPIP2;1 channel or overall transport activity due to the stabilization of a hydrophobic blockage inside the non-transporting SoPIP2;1 conformation. Literature focused on membrane protein biotechnology applications that also used SoPIP2;1 as a model system support the existence of hydrophobic blockages for non-transporting cases. Circular dichroism experiments have shown SoPIP2;1 to lose alpha-helical content and partially unfold when reconstituted into vesicle or liposome bilayers stiffened by cholesterol[59,60]. In fact, SoPIP2;1 structurally responds to these stiffer environments with hydrophobic movements within and between alpha-helices[60]. Stopped flow experiments where mercury was used to alter membrane fluidity have shown that SoPIP2;1 transport is affected by bilayer properties[61]. Mechanosensitive bias against cholesterol by SoPIP2;1 has been further validated using fluorescence experiments, where SoPIP2;1 preferably localizes in cholesterol-poor domains[62].

A general belief exists within literature that certain bilayer compositions can push equilibrium toward favorable conditions for membrane proteins. While cells can rapidly replace their lipid bilayer compositions, molecular models and simulations experience fixed stoichiometry. However, in silico simulated lipid molecules sample biophysical properties for direct modulation of membrane protein activity. Hindered transport of the open SoPIP2;1 in the complex bilayer is explained by the high negative mismatch in the height between the protein thickness and bilayer, as well as the complex bilayer composition's oversaturating level of sterols which leads to raft or crystal formation[47,48]. Specifically, sterols addition restricts phospholipid acyl tail conformational dynamics, increasing their order parameter and stiffness[63]. Indeed, reduction in water transport as a consequence of sterol addition was also reported in experiments involving AQP0[56] and AQP4[57]. For homogeneous bilayers, a decrease in the transport activity of the open state was associated with higher membrane stiffness (higher lipid order parameter) with channels transporting more than 30 waters per 100 ns. Additionally, rigid membranes of more than 0.19 in average order parameter cannot have high transport activity (more than 20 waters per 100 ns). Therefore, we can conclude that flexible membranes allow for the protein to sample more structurally and functionally diverse conformations.

Overall, we showed how a lipid bilayer can influence a membrane protein, leading to varying effects in structure, dynamics, and functions. Therefore, the selection of a membrane bilayer for MD simulation of a protein should be treated with care to prevent misinterpretation of results. SoPIP2;1, an AQP protein without specific lipid interactions, was modified in various ways by insertion into homogeneous and complex membrane bilayers. That is, lipid ensemble effects can lead to strikingly different observations as a result of bilayer embeddings. This phenomenon manifests in SoPIP2;1 by the stabilization, or trapping, of non-functional states, which leads to nearly no water transport across macrostates. It is possible that this phenomenon could occur *in planta* as a form of functional regulation. The concept of an "adaptive membrane" that selectively modulates membrane protein function and free energy landscapes has been discussed[8,41,64], although the lifetime of the ensuing protein regulation depends on the type of lipid-protein interaction[8]. Even so, the diversity and large quantity of lipids (along with the presence of other proteins) encompassing the membrane bilayer of a natural cell would induce diffusion and constant modification of the local lipid environment surrounding the protein as a countermeasure. Thus, the highly dynamic nature of the cell could prevent above-average sampling of any non-functional states observed from our SoPIP2;1 simulations. This, however, is a fundamental shortcoming of most classical MD ensembles employing a constant number of particles. This realization can also apply to in vitro experiments where lipid reconstitution, liposome, or nanodisc preparation occur with pre-defined lipid stoichiometries.

The field of protein-lipid modeling and simulation is shifting toward the usage of complex bilayers. As such, care is crucial when constructing a membrane bilayer model. As the relationship between lipid ensemble effects and protein conformational sampling or function remains a difficult research question to tease apart, MD

practitioners should thoroughly investigate known lipid-protein interaction for the system of interest to make the best system construction decisions[65,66]. Proteins requiring specific interactions with lipids depend on those high-affinity binding events (e.g., sterols with GPCRs). Unlike more common transient/non-specific or somewhat specific interactions, high-affinity and specific binding interactions are long-lived events occurring on the microsecond timescale[8]. Under equilibrium conditions, these specific interactions should energetically dominate over lipid ensemble effects that would otherwise lead to compensatory deformations along the modeled membrane surface, such as changes in curvature, thickness, or rigidity/fluidity. Given that the recruitment of specific lipids is a known phenomenon[10], criteria for including lipid species with a high affinity and/or specific interaction with the protein should be satisfied during system construction. Knowledge on functionally relevant lipids should be retained and used for the design of in vitro or model membrane reconstitution systems.

However, preferences for specific lipid species are not known for all membrane proteins. If specific protein-lipid interactions are known for a homologous protein, these interactions could potentially be adapted to the target modeled protein for simulation. Otherwise, it can be assumed that a bilayer composition that approximates the native cellular environment should provide lipid chemistry that can directly offer, or mimic, the unknown specific interactions. To this end, greater care must be applied when performing simulations with a realistic bilayer, as the annular shell configuration used to seed the simulation could inhibit phase space exploration. At a minimum, a composition that represents some average of the ensemble properties of a more complex or realistic bilayer should be used.

One should also consider employing replicates with varied membrane packings to overcome the possibility of skewed samplings. Even when starting with different initial membrane packings for the open and closed states, non-productive SoPIP2;1 states were still observed in this study. Asymmetric sterol distribution is necessary for biological bilayer integrity, and realistic bilayers possess tightly packed aggregates of sterols[58]. The objective of the MD practitioner now becomes equipping membrane protein simulations with a realistic, or appropriate, bilayer choice that enables functionally-relevant discovery while respecting real-life biological constraints. Our conclusion is especially important for researchers running short membrane protein simulations, as initial membrane packings are not likely to significantly deviate along the nanosecond timescale. This means that our lipid-induced SoPIP2;1 inhibition does indeed happen in cells, and that efforts must be made to also observe functional dynamics during simulation. Membrane mixing offers a solution and computational analogy to lipid imbalance regulation performed by flippases and scramblases[67]. Membrane mixing will also redistribute chemically identical, but biophysically distinct, lipid molecules, which may alleviate rigidity within annular shell arrangements (Fig. 8). Tools for altering membrane configuration have since become widely available to further diversify membrane packings[13,68,69].

For guiding enhanced sampling protocols and kinetic features, we recommend MD practitioners apply a combination of lipid and protein features. For path sampling techniques, lipid acyl chain order parameters could be used as an additional reaction coordinate to ensure that structurally desired protein seed structures or windows are operating within a membrane configuration that promotes functional states. Other examples of non-protein variables that confound protein conformational dynamics, like buried water molecules or cavity dewetting, have been previously identified in potassium channels[70,71].

Extensive molecular dynamics simulations have been performed on plant membrane transporters[22–24] and hormone receptors[72,73]. However, simulation or experimental investigations of how plant membrane bilayers impact proteins conformational dynamics are limited. The observations from this study contribute to a better understanding of plant aquaporin dynamics and functions under lipid influences. Tuning

of the local lipid environment near aquaporin can enhance the sampling of desired conformations or functional states. As aquaporins are found to be crucial in human disease proliferation or *in planta* water regulation, aquaporin regulation can contribute to drug intervention or water conservation, respectively. For instance, an increased sterol level near the protein can lead to the non-functional open state being more frequently sampled, preventing water transport without incurring energetic costs related to changing loop D conformation.

These insights on how selective lipid environments can exert spatiotemporal control on membrane proteins could be coopted for engineering efforts involving membrane proteins. As aquaporin has recently been employed in proprietary biocompatible water filtration system (NASDAQ: AQP), any synthetic membrane encompassing an aquaporin protein could theoretically be adjusted to have a local lipid environment that induces high water transport, leading to increased filtration efficiency. The inverse case is also possible, where membrane properties could be selectively tuned to create non-functional aquaporins without incurring energetic costs associated with cytosolic loop closure. At the very least, insights from our work can be directly applied to the study of water stress responses through plant aquaporin function, an area for which a demand exists for dedicated molecular simulation studies[74]. Aquaporin was chosen as a model system to uncover unspecific findings around lipid-protein interactions that could be generalized to other membrane protein classes. However, more studies and experimental validations are required to employ these possible applications of lipid-protein interactions on aquaporins, and to extend these findings to the dynamical characterization of all membrane proteins.

## Methods
### System assembly
Starting coordinates for the simulation were acquired from the Protein Data Bank (PDB), including the SoPIP2;1 open (PDB ID: 2B5F) and closed (PDB ID: 1Z98) crystal structures[21]. SoPIP2;1, and aquaporins in general, have a conserved topology of 6 transmembrane domain (TM) and 2 half-helices spanning the upper (TME) and lower leaflet (TMB) of the membrane (Supplementary Fig. 1). Monomer chain A of the open and closed tetrameric crystal structures were each obtained and aligned to contain residues 28 to 263 as to have the same length. The $Cd^{2+}$ ion located near the N-terminus of the closed crystal structure was replaced with $Ca^{2+}$ to be representative of living systems[21]. The open and closed monomers were aligned, and the $Ca^{2+}$ in the closed conformation was copied to the open conformation. The crystal structure starting states contain the $Ca^{2+}$ ion near the N-terminus. Charged states of titratable residues were predicted with the PDB2PQR web server[75] with the PROPKA[76] option. Missing hydrogen atoms were filled during system-building through CHARMM-GUI[77].

The computational efficiency associated with the use of monomer SoPIP2;1 is further justified by experimental findings and related literature reviews. AQP monomers are known to constitute functionally independent pores[28]. Each AQP monomer can be functionally reconstituted in vitro[78,79], and even functional AQP monomer structures have been resolved by solid state NMR spectroscopy (PDB IDs: 8H1D and 6POJ)[80,81].

Previous experiments determined the phosphorylation of Ser115 and Ser274 in SoPIP2;1 to play a role in maintaining an open channel, and the dephosphorylation process induces the closed conformation[82,83]. Specifically, in the presence of a dephosphorylation inhibitor, the channel's water transport activity increased significantly compared to the positive control[82]. Thus, if Ser115 and Ser274 are phosphorylated in classical MD (no bond breaking/forming allowed), the cytosolic loop could remain constitutively open, and the closing of the open channel might not be observed in the microsecond timescale of MD simulations. This observation suggested that the phosphorylation/dephosphorylation process provides thermodynamic stability for

one conformation over the other. When phosphorylation is inhibited, SoPIP2;1 can still sample both the open and closed states, as water transport activity is only reduced rather than altogether abolished[82]. In this case, the barrier of loop conformational change for SoPIP2;1 is assumed to be mostly kinetic. Therefore, Ser115 and Ser274 were not phosphorylated in this study to ensure sampling of both the opening and closing of the channel.

The systems were prepared in CHARMM-GUI for simulations[77,84,85]. Nine homogeneous bilayer systems containing POPC (1-palmitoyl-2-oleoyl-sn-glycero-3-phosphocholine; 16:0/18:1), POPE (1-palmitoyl-2-oleoyl-sn-glycero-3-phosphoethanolamine; 16:0/18:1), POPG (1-palmitoyl-2-oleoyl-sn-glycero-3-phosphatidylglycerol; 16:0/18:1), PLPC (1-palmitoyl-2-linoleoyl-sn-glycero-3-phosphocholine; 16:0/18:2), PLPE (1-palmitoyl-2-linoleoyl-sn-glycero-3-phosphoethanolamine; 16:0/18:2), PLPG (1-palmitoyl-2-linoleoyl-sn-glycero-3-phosphatidylglycerol; 16:0/18:2), LLPC (1-linoleoyl-2-linolenoyl-sn-glycero-3-phosphocholine; 18:2/18:3), LLPE (1-linoleoyl-2-linolenoyl-sn-glycero-3-phosphoethanolamine; 18:2/18:3), LLPG (1-linoleoyl-2-linolenoyl-sn-glycero-3-phosphatidylglycerol; 18:2/18:3) and one heterogenous complex system[12] were built to contain 128 lipids. The lipid species composing the realistic bilayer were POPC, PLPC, PLPE, PLPG, LLPC, LLPE, DLiPC (1,2-dilinoleoyl-sn-glycero-3-phosphocholine; 18:2/18:2), DLiPE (1,2-dilinoleoyl-sn-glycero-3-phosphoethanolamine; 18:2/18:2), STIG (stigmasterol), and SITO (ß-sitosterol). Specific compositions of each lipid species in the complex system can be found in Fig. 1b. The homogeneous bilayers were constructed to be symmetric, with 64 lipids per leaflet. Meanwhile, the complex bilayer was asymmetric, with 66 lipids in the upper leaflet and 62 lipids in the lower leaflet. ACE/CT3 terminal patchings were used for SoPIP2;1. The systems were solvated with TIP3P waters[86] and neutralized with 0.2 M CaCl$_2$. Upper and lower water layers were 15 Å in height. Force field parameters were CHARMM36[87] with existing pair-specific NBFIX (nonbonded fix) Lennard-Jones parameters for Ca$^{2+}$ and Cl$^-$ ion pairing, as well as Ca$^{2+}$ and phosphate group-oxygen atom pairings[88–90]. Hydrogen Mass Repartitioning (HMR, discussed in Simulation Details) was used to increase the timestep from 2 fs to 4 fs[91]. Using HMR to maximize the timestep contributes to longer trajectories under fewer computing resources, thereby improving conformational sampling and kinetic calculations for protein conformational changes.

## Simulation details

Classical molecular dynamics were performed using AMBER18[92]. Each system underwent energy minimization, NPT heating, NPT "hold" with constraints, and equilibration[93]. Specifically, during energy minimization, the steepest descent and conjugate gradient algorithm were used with a maximum of 50,000 iterations, in which the first 5000 steps utilized the steepest descent algorithm. Then, each energy-minimized system was heated from 0 K to 300 K in an NPT ensemble for 2 ns with *sander*. A 5 ns NPT hold was done at 300 K and 1.13 bar under *pmemd*, in which the protein backbone C$_\alpha$ atoms were constrained by a spring force with constant 10 kcal mol$^{-1}$ Å$^{-2}$. Afterward, an equilibration run with no protein constraints was performed for 10 ns on *cuda*. Production runs were subjected to Langevin thermostat[94] and Monte Carlo barostat to maintain a constant temperature of 300 K and a constant pressure of 1.13 bar. The Verlet integrator was used with a timestep of 4 fs. Hydrogens are the lightest atoms in the systems, having vibrational frequency much lower than the femtosecond timestep of MD simulations, which can lead to large fluctuations in the integration step. To minimize this drawback, the SHAKE algorithm[95] was implemented to constrain the non-water hydrogen atoms along with HMR to distribute neighboring non-water heavy atoms' mass to hydrogen atoms.

For nonbonded interactions, the Lennard-Jones cutoff was set at 12 Å. For long-range electrostatic interactions, Particle Mesh Ewald was used[96]. Periodic boundary conditions were maintained during simulations. To evaluate water transport in the SoPIP2;1 cavity, the frame save rate was chosen at 10 ps, as simulations of water transport in AQP1[32] showed the fastest significant water motion inside the pore to occur in 10 ps. Additionally, we tested the frame save rates of 5 ps, 10 ps, and 30 ps in a SoPIP2:POPG equilibration run of the same 10 ns trajectory and visualized the water movements for each frame save rate. We found that the Cartesian coordinates of waters in the 10 ps frame save rate were captured with a similar resolution to those from the 5 ps frame save rate run, and the 30 ps rate run produced too large a fluctuation in water movement. Thus, a 10 ps frame save rate offers the best compromise of accurately recording water dynamics without creating unreasonably sized trajectory files.

## Adaptive sampling

After an initial 1 μs of classical MD from the open and closed systems, an enhanced sampling method known as adaptive sampling[97–102] was implemented. Adaptive sampling consists of running multiple short unbiased trajectories in parallel to reduce computing time and enhance sampling of the free energy landscape, compared to running one long trajectory. This method incorporates the selection of "seeding frames" for the subsequent round of trajectories based on the free energy landscape from the previous round, improving the sampling of rare states in high-energy conformations that are difficult to capture in long continuous simulations. Twelve pairs of distances identified as the most different between the closed and open crystal structures were used as an initial basis for adaptive sampling. The twelve distances underwent dimensionality reduction with time-lagged independent component analysis (tICA)[103]. MDTraj 1.9.4 was used for the calculations of the distances[104]. The energy landscape projected onto the first two principal components identified by tICA was constructed using PyEMMA 2.5.6 to visualize the transition between the open and closed macrostates[105]. Frames with the highest free energy that were in the transitionary region between the open and closed macrostates were selected from the tIC space. Then, the k-means clustering algorithm (scikit-learn 0.21.2) was implemented to select 10–20 seeds belonging to the least populated cluster for the next round of simulations. Each selected seeding frame then underwent 100 ns of classical MD. This workflow was repeated until the tIC energy landscape showed connectivity between the open and closed macrostates, thereby indicating the reversible opening-closing transition of SoPIP2;1. The total simulation time for each system ranged from 19–55 μs, and the specific time was listed in Supplementary Tables 1–11. In total, we performed 315.69 μs of classical MD for this study.

## Data-driven feature selection and Markov state models

The Markov state model (MSM) is a theoretical framework developed for understanding the thermodynamics and kinetics of a biomolecular system given numerous trajectories generated from enhanced sampling workflows[103,106–109]. Markovianity applies to data from MD simulations because the computation of an atom's position and velocity depends only on its immediate previous position and velocity. MSM construction generates a transition count matrix containing the raw counts of the transitions among microstates. The microstates are approximated from k-means clustering on the selected geometric distance features of all sampled protein conformations (all frames in the simulations). The transition between these microstates is counted after a specific lag time, which corresponds to the time after which the Markovian detailed balance is satisfied. Then, the probability of transition can be obtained from the counts. From the Markov transition probability matrix, the stationary distribution of each microstate can be calculated as the first eigenvector of the matrix. Then, the microstates closest to the crystal structures are identified by minimizing the Euclidean norm of the difference between the microstate and the crystal structure distance features. The mean first passage time (MFPT) for the system to transition from the open microstate to the closed

microstate (and vice versa) can be calculated as the inverse transition rate. MSM generation includes featurization, dimensionality reduction, clustering, and hyperparameter optimization. We performed a data-driven approach for the first step, featurization, where a metric must be selected to characterize the protein.

The twelve distances used for dimensionality reduction in the adaptive sampling workflow did not result in reasonable MSM construction based on validation tests. An alternative feature selection protocol was performed to enable proper phase space discretization. A residue-residue contact scoring (RRCS)[110] method was implemented in combination with spectral oASIS[111]. The residue-residue distances of the protein (excluding the first five residues in the N-terminus, due to their extensive movement as a random coil) in each frame were subtracted from the closed and open crystal structure residue-residue distances and normalized across each frame. The distribution of distances was analyzed to extract distances outside the ±1.5 $z$-score region, resulting in distances that varied the most from the crystal structures during simulations. The obtained 1500–3000 features then underwent spectral oASIS[111], implemented as part of PyEMMA 2.5.6[105], in which a Nyström matrix operation performs an automatic selection of feature subsets that can approximate the leading eigenvalues and corresponding eigenvectors of the time-lagged covariance matrix of the original feature set. Spectral oASIS requires a pre-specification of the number of final features. Thus, a grid-searching protocol was implemented to search for 20–50 final features, with increments of 5, to maximize the VAMP-2 score (variational approach to Markov processes)[112]. If the transition probability of the microstates follows the Markovian detailed balance, the VAMP-2 score equals the sum of the matrix's highest eigenvalues squared. Therefore, to achieve this goal as closely as possible, heuristic variables for MSM construction should allow for the maximization of the VAMP-2 score. The protocol was repeated for each system to ensure unbiased, completely data-driven identification of system-dependent thermodynamics and kinetics information extracted from MSMs. The number of final features is presented in Supplementary Table 1, and the residues in each feature are listed in Supplementary Fig. 2. Because SoPIP2;1 dynamics deviated between simulations performed with different bilayer constructs, resulting hyperparameters and discretization feature set were independently optimized for each SoPIP2:bilayer system. In this way, acquired transition probability matrices will be calculated based on the most accurate representation of SoPIP2;1 phase space when simulated with a specific bilayer.

Utilizing the respectively selected feature sets, MSMs were built for each system using PyEMMA 2.5.6[105]. To optimize hyperparameters for MSM construction, a grid search was done to maximize the VAMP-2 score (explained in the preceding paragraph) of the resulting MSMs. tIC dimensions were varied from 4–12 for every system. The number of clusters varied from 300–1000, with 100 increments. For the grid search, tICA lag time was maintained at 100 ps, and MSM lag time was 2 ns. Hyperparameters that produced the highest five VAMP-2 scores were further investigated. With the implied timescale plot, the MSM lag time was selected to be the fastest time at which the implied timescale converges. The final MSM was built with the selected lag time to produce the equilibrium distribution of the microstates. The parameter sets with the most converged implied timescale plot (Supplementary Fig. 3) and an equilibrium distribution that did not deviate above 1.5 orders of magnitude from the original distribution (Supplementary Fig. 4) were selected. The final parameters are listed in Supplementary Table 1. The Chapman-Kolmogorov tests were performed to validate built MSMs and ensure stochastic processes' detailed balance requirement (Supplementary Figs. 5–14). The mean first passage time between the open and closed crystal structure on the free energy landscape was also calculated with PyEMMA 2.5.6[105].

Sampling errors were quantified by bootstrapping with 200 data subsets, each containing 80% of the total data. Each bootstrapping sample was randomly selected as ~80% of the original MSM clusters. A new MSM was then built for each sample. The MFPT between the open and closed crystal structures was computed for each sample, and the MFPT error of the full MSM was reported as the standard error of the mean of the 200 MFPT sample values. A binning protocol was followed to calculate and project the free energy errors from these sample MSMs onto the original principal tICs[22,23]. Specifically, data was grouped into a 2D histogram with edges defined by the lower and upper bound of the original tICs. The free energy error was computed for each histogram bin as the normalized standard deviation of the relative energy across the 200 samples (Supplementary Fig. 15). Links for the scripts used for the calculations and validations related to MSM can be found in the Supplementary Information.

### Trajectory selection and analysis

Frames in the minima of the free energy landscape projected onto the first two tICs were randomly sampled for 1000 discrete frames. The representative frames for each bilayer-SoPIP2;1 system were used for the structural analyses of the protein and lipid bilayer, including hydrophobic mismatch and Leu197 dihedral. These observables were reported as distributions. Because the water transport function of SoPIP2;1 requires a continuous trajectory for analysis, three independent trajectories with the most frames that lie in each minimum of the free energy landscape were selected. The trajectories were visualized on the free energy landscapes in Supplementary Fig. 17. Errors were calculated as standard errors of the mean among these three trajectories or among the trajectories of the same macrostate. To shed light on structural information of the protein pore that influences water transport, HOLE analysis[113] was also performed on continuous trajectories for each macrostate of each bilayer-protein system.

### Structural analysis

The discrete frames from each minimum underwent analysis of "pore plug" Leu197's (connecting loop D and TM4) dihedral angle with the first residue of the opposing helix (TM5), Ala182. MDTraj 1.9.4[104] was used for the calculations of the dihedral angle. Output values were represented as $n\pi$. The four atoms involved in the calculation are visualized in Fig. 4b.

### Functional analysis

To understand the water transport activity of SoPIP2;1 in each system, we computed the number of waters imported/exported and the rate at which each water molecule was transported by adapting scripts from Gelenter et al.[30] using MDAnalysis 2.0.0[114]. HOLE radius[113] of representative non-transport/transport trajectories and the z-position distributions of the $C_\beta$ of important residues inside the channel were also extracted from the trajectories with MDAnalysis 2.0.0[114].

### Lipid binding site analysis

Generally, AQP proteins are known to have conserved lipid binding sites. To compare how lipid binding occurred throughout SoPIP2;1 loop D opening/closing transitions, PyLipID software was used to calculate lipid-protein residue interactions[115]. The default lipid contact distance cutoff parameters of 4 to 8 Å were used as suggested by PyLipID tutorials. Calculated results were averaged with residence times reported in nanoseconds and presented as heatmaps. For SoPIP2:bilayer embeddings where more than one metastable intermediate state existed, the intermediate state residence times were averaged to determine the resulting heatmap. For reporting state-specific loop D interaction results, radial bar graphs were used to generate "fingerprints" with the resulting PyLipID residence time data. Because the SoPIP2:complex simulations involve different lipid species, the maximum SoPIP2:lipid interaction residence time was reported for each residue.

After the binding site calculations were performed on continuous trajectories representing each of the metastable states, the results

were compared to generally observed trends in previously reported AQP lipid binding site analyses. Specifically, we compared against results from multiscale simulations performed by Stansfeld, Jefferys and Sansom[39], as well as the MemProtMD database[40]. We first retrieved each AQP tetrameric structure from the MemProtMD database except for PDB IDs 2B5F and 1Z98, totaling 52 unique AQP PDB IDs. Under visual inspection, each of the 52 tetramers was divided into monomeric units. Because our SoPIP2;1 simulations were performed in the monomeric form, we focused our analysis by only focusing on potential lipid binding sites that would be seen under the SoPIP2;1 tetramer assembly. Using the MemProtMD 2B5F and 1Z98 models, we identified four potential lipid binding sites based off lipid-facing orientation. These four lipid-exposed stretches on SoPIP2;1 included (1) Asp28 to Tyr53, (2) Phe86 to Gln147, (3) Gly158 to Phe204, and (4) Pro223 to Val263.

With the putative SoPIP2;1 tetramer binding sites identified, each monomeric protomer separated from the 52 MemProtMD tetramers was structurally aligned to a SoPIP2;1 monomer. After structural alignment, multiple sequence alignments were generated using only structurally aligned MemProtMD monomer PDB files as input for Promals3D[116]. Resulting multiple sequence alignments were analyzed to determine where SoPIP2;1-homologous lipid-exposed stretches existed on each MemProtMD monomer. These sequence mappings were visually confirmed for each of the 208 monomer structures. Once the sequence mappings were structurally confirmed, the maximum of the MemProtMD lipid head and tail contact probabilities were retained for each monomer residue. The resulting lipid contact probability for each structurally equivalent residue was averaged, yielding a singular per-residue value for each of the 52 MemProtMD structures.

To compare our simulations results against the MemProtMD structures, SoPIP2;1 per-residue lipid PyLipID residence times were averaged across all representative open, intermediate, and closed state trajectories and then reported as a contact probability.

### Lipid bilayer analysis

For the hydrophobic mismatch calculation of the selected 1000 frames in each minimum, Membrainy 2021.2[117]. was used. Membrainy provided the coordinates of the annular shell, the thickness of the annular shell, and the thickness of the whole membrane[117]. We used the subtraction of the average height of phosphate groups in each leaflet for the membrane thickness calculation. The maximum absolute $z$ values across six transmembrane helices were subtracted to obtain the protein thickness. Then, the difference between the protein and bulk membrane thickness was compared with the difference between the annular shell and the bulk membrane thickness. Bilayer lipids encapsulating the membrane protein are crucial in influencing the functions of some AQPs, as seen for the transport of water by AQP0[56] and AQP4[57], despite AQPs having no specific binding site with lipids. Therefore, we analyzed the lipid order parameters of each lipid molecule in the membrane for the continuous trajectories. LiPyphilic 0.10.0[55] was used to calculate the order parameter of each lipid at each frame. The thickness of the membrane through the continuous trajectories was calculated with Membrainy 2021.2[117]. Area-per-lipid (APL) calculations for the initial 1 μs production trajectories were also calculated using Membrainy to test for convergence (Supplementary Fig. 18).

### Trajectory file processing

Trajectory files were analyzed and processed by AmberTools CPPTRAJ 6.4.4[118], including conversion from *.nc* to *.xtc*, stripping of lipid and/or water molecules for faster analysis calculations, and selecting frames for analysis.

### Figure generation

Numerical figures were plotted with the Matplotlib 3.2.2, Seaborn 0.11.2, and Plotly 5.13.0 packages on Python 3.6. Panel figures and graphics were generated with Adobe Illustrator 2023. Snapshots of the simulations, including the proteins, waters, and lipids, were generated with UCSF Chimera[119].

### Reporting summary

Further information on research design is available in the Nature Portfolio Reporting Summary linked to this article.

## Data availability

The data that support this study are available from the corresponding authors upon request. Trajectory files, MSM objects, and analysis pickle files are available on Box, which can be found in GitHub [https://github.com/ShuklaGroup/Lipid_composition_on_AQP]. Individual trajectory files upon request. Otherwise, representative trajectories and frames used for discrete or continuous trajectory analyses, along with the relevant Source Data generated by analysis calculations can be accessed using links made available through our Github [https://github.com/ShuklaGroup/Lipid_composition_on_AQP] and Dryad [10.5061/dryad.jsxksn0hc. 2024] repositories. Source Data needed to reproduce the figures in this manuscript has been uploaded to our Dryad repository. The following previously published PDB accession codes were referenced in the Main Text of this document: 2B5F; 1Z98; 8H1D; 6POJ.

## Code availability

All code and related environments used for analyses are detailed within our GitHub repository: https://github.com/ShuklaGroup/Lipid_composition_on_AQP.

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

## Acknowledgements

D.S., A.T.W. and A.T.P.N. acknowledge funding from the National Institutes of Health (Award No. R35GM142745). A.T.P.N. acknowledges the Undergraduate Research Fellowship from the Beckman Institute for Advanced Science and Technology, the Preble Research Award from the James Scholar Honors Program at the College of Liberal Arts & Sciences, and the John A. Weedman Scholarship from the Department of Chemical and Biomolecular Engineering at the University of Illinois Urbana-Champaign. The authors thank Soumajit Dutta for useful discussion regarding Markov state model theory and construction. The authors thank Troy Brier for lending of a Linux machine to help facilitate analysis calculations and equilibration runs. The authors also thank Krishna Narayanan for comments on earlier drafts of this manuscript.

## Author contributions

ATW conceived and designed the project. ATPN performed simulations with some help in system construction from ATW. ATPN wrote code, managed the data, and performed all analyses. ATW performed APL and lipid binding analyses. ATPN interpreted findings and developed additional analyses with feedback from ATW and DS. ATPN and ATW wrote the manuscript with input from DS. ATW and DS handled text revisions and the rebuttal letter. ATW helped manage project progress through direct reporting from ATPN. DS supervised the project.

## Competing interests

The authors declare no competing interests.
