## [Peer Review File · Nature Communications]

Functional Regulation of Aquaporin Dynamics by Lipid Bilayer CompositionReviewer #1 (Remarks to the Author):

In this study, Nguyen et al use MD simulations to investigate the impact of lipid composition on the dynamics and function of a model protein, an AQP. After careful analysis, the authors observe that the membrane composition impacts the AQP in a range of ways, including thermodynamical, kinetic, and functional. Their findings feed into an ongoing conversation about the choice of lipids in MD simulations of membrane proteins. As such, it is an important study, with widely interesting and useful results.

The study is clearly well carried-out, and the ms is very thorough and well written. The sampling is extensive, the techniques are rigorously applied, and a range of appropriate analysis methods have been applied (tICA, MSMs etc). In all, I found the study to be very high quality.

I support publication, pending a few small changes:

- I feel the main take-home messages of the ms get a little lost in the analyses. To aid the stretched-for-time reader, it might help to make sure the key messages are properly highlighted up front, i.e. that a single lipid can give unusual effects (esp POPC, which is a very common lipid in MD sims), that complex membrane has things to watch out for, etc. i.e. what advice do the authors give for the community?
- It's mentioned that, based on the simulation data, crystal structures might not represent the most stable conformations in lipid bilayers. However, the analysis has also been performed on monomeric AQP, whereas I believe the AQP xtal structures were tetramers? This might partly explain this phenomenon. It would be nice to see a single repeat (e.g. with an "unstable" lipid, like POPE) done for the tetramer if possible.
- Are the simulations long enough to allow the membrane to equilibrate? Esp for the complex membrane, it would be good to make sure equilibration has occurred, as 1 μ s might not be long enough for this. This might partially explain things like the hydrophobic mismatch.
- As a limitation/future direction, it's worth discussing the role of asymmetry in model membranes, as discussed in a recent preprint, which suggests a strong asymmetry of PLs and sterols in the membrane (10.1101/2023.07.30.551157).
- In Fig 3A – the "closed" transporter transports more water than "open" for POPE, POPC, PLPC, LLPG. Does this suggest that these lipids are giving qualitatively incorrect data?

Minor:

- Fig 1B – the lipids are a bit too small to see.
- Same for the labels in 1C (maybe use a shared axis?). Can the name of the lipid be added in large print on the graph? Probably instead of the titles.
- On 1c, the red and blue dots could be clearer – maybe bigger and not a colour used in the heatmap? (i.e. black or white perhaps)
- 2c and 3b labels are too small

Robin Corey

Reviewer #2 (Remarks to the Author):

Review of NCOMMS-23-33126

This is an interesting manuscript using molecular dynamics (MD) simulations to explore the possible functional regulation of a plant Aquaporin (SoPIP2;1, henceforth SoPIP for short) by membrane lipids. As noted in the abstract this Aqp has no reported high-affinity lipid interactions, although this may

reflect e.g. the age of the SoPIP X-ray structures – lipid like density has been observed in a number of membrane protein structures determined recently by cryo-EM including e.g. AQP2. Furthermore, older simulations have been used to explore Aqp/phospholipid interactions (see e.g. doi: 10.1016/j.str.2013.03.005) and have been shown to reveal lipid binding sites seen in a handful of early high resolution structures of Aqp0 and Aqp4. Indeed, it is puzzling that this earlier literature is not reviewed in the Introduction to the current ms. Having said that, SoPIP is an interesting candidate for further investigation of Aqp/lipid interactions, given that it shows regulation (gating) of water flow by the conformation of a cytosolic loop which is not seen in other Aqps. The authors therefore focus on possible lipid effects on such regulation.

All atom simulations were performed for SoPIP in nine different single lipid bilayers and one more complex mixed lipid bilayer. The simulations were in the presence of 0.2 M Ca ions. To what extent are we confident that the CHARMM36 forcefield reproduces lipid/Ca ion interactions accurately? I am aware of several simulation studies attempting to improve Ca ion interactions with lipids. Multi-microsecond simulations, adaptive state sampling and Markov state models were employed in the simulations. This is a careful, state of the art approach. Analysis focussed on the functional dynamics of the protein, especially the pore plug, and effects on water transport.

I have two main questions which need to be addressed.

1. Are there any experimental data for lipid effects on SoPIP function which correlate with the simulation results?

2. Was any analysis of SoPIP/lipid interactions performed? Are any lipid binding sites comparable to those for other Aqps (see comment above) or for other membrane proteins observed?

Taken together these are important as the lipid analysis in the manuscript concentrate on bilayer biophysical properties e.g. possible hydrophobic mismatch. The authors conclude (page 28) that “the lipids used in this study can adapt to SoPIP_{2;1}”. I am perhaps not surprised that the local thickness of a bilayer adapts to the structure of a relatively rigid membrane protein such as an Aqp, rather than vice versa (as might be expected for a protein which could adopt different conformations within a bilayer, e.g. an ion channel or a transporter). Given the regulation of SoPIP involves a cytosolic loop I would have expected a more detailed examination of specific loop/lipid interactions. The authors state that “Overall, the transition path from closed to open varies drastically for the same protein embedded in different membrane bilayers”. Given this I would have expected at least some search for specific lipid interactions. Even if these were not observed, they could then be ruled out as a mechanism, thus refocussing attention on bilayer biophysical properties such as local thickness.

Please find versions of our revised and annotated manuscript main text, as well as our supplementary information, attached. Below, we feel we have addressed the reviewer comments posted during the peer review process. We respond to each concern in a point-by-point manner. We highlight new text, verbatim, using a blue font. Previous text is written using a red font. Discussion is otherwise maintained with a default black font color.

REVIEWER COMMENTS

Reviewer #1 (Remarks to the Author)

In this study, Nguyen et al use MD simulations to investigate the impact of lipid composition on the dynamics and function of a model protein, an AQP. After careful analysis, the authors observe that the membrane composition impacts the AQP in a range of ways, including thermodynamical, kinetic, and functional. Their findings feed into an ongoing conversation about the choice of lipids in MD simulations of membrane proteins. As such, it is an important study, with widely interesting and useful results.

The study is clearly well carried-out, and the ms is very thorough and well-written. The sampling is extensive, the techniques are rigorously applied, and a range of appropriate analysis methods have been applied (tICA, MSMs etc). In all, I found the study to be very high quality.

We thank you for your review and are glad that we were able to get your perspective on our work, given your expertise and publication record in the field of protein-lipid interactions. The writing of this manuscript was a bit of a surprise, as we initially assumed more straightforward trends according to degree of acyl tail unsaturation and headgroup choice. You are right in your comments that the field deserves more straightforward delivery on some of our conclusions and streamlined advice. We provide such scientific commentary below.

I support publication, pending a few small changes:

- **I feel the take-home messages of the ms get a little lost in the analyses. To aid the stretched-for-time reader, it might help to make sure the key messages are properly highlighted up front (esp POPC, which is a very common lipid in MD sims), that complex membrane has things to watch out for, etc. i.e. what advice do the authors give for the community?**

We agree with this take. Writing this manuscript was a little tricky, as we wanted to tailor it towards a broad scientific audience. We desired to provide a detailed discussion on how lipid bilayer environment functionally regulated SoPIP₂,;1 and at times the take-home messages do

become a bit buried. Making the take-home messages more streamlined for cursory readers will benefit all that are interested in this manuscript.

We have decided to add a summary paragraph to conclude the Introduction section. Our take-home points are presented as follows on Page 4:

“To summarize our key take-home points, we find that membrane choice induces different slowest processes, which will inherently alter thermodynamics, kinetics, and functional observations. However, more than one model bilayer can appropriately model target membrane protein function. We recommend that MD practitioners research known lipid-protein interactions to make the best system construction decisions. Ensemble average properties revealed from literature search or lipidomic data from a related organism/cell type should be represented in the modeled bilayer. The configuration of the annular shell(s) used to seed simulations should be diversified. MD practitioners should employ replicates with varied membrane packings to avoid artifacts caused by initialized configurations for both simple and complex bilayers. Realistic bilayers with asymmetric sterol distributions could cause tight bilayer packings that could drastically affect results. Lastly, we encourage researchers to confirm that computationally observed states are functional. Complementary analyses should be used to build trust in observed structures. When applying enhanced sampling techniques, caution must be exercised by inspecting starting states or using a combination of lipid bilayer and protein features to drive simulations.”

- **It's mentions that, based on the simulation data, crystal structures might not represent the most stable conformations in the lipid bilayers. However, the analysis has also been performed on monomeric AQP, whereas I believe the AQP xtal structures were tetramers? This might partly explain this phenomenon. It would be nice to see a single repeat (e.g., with an “unstable” lipid, like POPE) done for the tetramer if possible.**

We wanted to reproduce a opening/closing free energy landscape using a tetramer structure, but applying the adaptive sampling protocol to a system with over 100k atoms could take up to a year. We could have obtained short simulations of the tetrameric SoPIP2;1 in different bilayers, but these would not adequately provide the most stable conformation within the different metastable minima. Additionally, the construction of tetramer systems would become complicated by accounting for cooperativity between individual protomers.

However, the raised concern is a great point. To meet the four-week deadline for peer review, we conducted a useful literature review concerning AQP structure and function. Within cells, AQP proteins are ubiquitously found as tetramers. However, it is also known that each monomer constitutes a functionally independent pore (Ozu *et al.*, 2022). Within this study, we chose to simulate monomeric SoPIP2;1 due to the increased

computational efficiency. Although less common, functional aquaporin monomer structures have been resolved by solid state NMR spectroscopy (PDB IDs: 8H1D; Tan *et al.* 2022) and 6POJ (Dingwell, Brown & Ladizhansky, 2019). Each AQP monomer can indeed be functionally reconstituted *in vitro* (Borgnia *et al.*, 1999; Schmidt & Sturgis, 2017). Still, in nature, tetramerization is mandatory in AQPs (Ozu *et al.*, 2022).

Exoplasmic loops along some of the monomers directly engage in hydrogen bonding contacts and/or disulfide bridges to help with monomer association and tetramer assembly (Gössweiner-Mohr *et al.*, 2022). While these exoplasmic loop contacts help with tetramer formation, they are not required to maintain tetrameric structure (Roche & Törnroth-Horsefield, 2017).

For plasma membrane intrinsic protein (PIP) aquaporins, like SoPIP2;1, loop A participates in tetramer formation. From our tICA decomposition analyses, a few systems actually demonstrated loop A as one of the slowest processes during loop D opening/closing transitions (original submission Figure 1; now Figure 2). These systems include POPE*, POPG, PLPE*, LLPE, LLPG*, and the complex bilayer*. While each of these bilayers offer significant distinction between the open and closed loop SoPIP2;1 states (except LLPE), all of them (except POPG) might not represent one of the crystal structures as one of the most stable conformations.

There is experimental evidence that suggests some AQP monomers to be less stable than AQP tetramers (Cymer & Schneider, 2010). Considering that the opening/closing transition in the above lipid embeddings is more dependent on loop A dynamics, we agree with you that these crystal structure poses would likely become more stable under tetramerization. Despite this reduced stability, the states are still functional. Our water transport analyses (original submission Figure 3; now Figure 5) show that POPG and LLPE – two systems where loop A dynamics becomes one of the slowest processes – do maintain expected functional behavior (i.e., open states conduct more water transport than closed states; closed states transport minimal to zero waters).

So overall, yes, we agree that simulation of a tetramer would likely stabilize some of the embeddings that present the crystal structure poses as “unstable” (i.e., those with loop A dynamics as one of the slowest processes). However, we cannot complete a tetramer simulation in the four weeks provided for resubmission of the revised manuscript. Instead, we can reflect the points of this peer-review discussion in the main text.

In response, we have made the following changes to the Results section “Aquaporin Conformational Dynamics”:

“Despite being well discretized, the crystal structure poses shown in the tICA landscapes for POPE, PLPE, LLPG, and the complex bilayer are not always the most stable conformations (Figure 2). Each of these SoPIP2:lipid embeddings share having loop A dynamics as one of the dominant tIC components. POPG and LLPE also are dominated by loop A dynamics, although POPG crystal structures are indeed the most stable. Conversely,

LLPE tIC decomposition places both crystal structures within the same minima. For plasma membrane intrinsic protein (PIP) aquaporins, like SoPIP2;1, loop A participates in tetramer formation.[26] Loop A, and its homologous sequences in non-PIP orthodox AQPs, typically engage in direct hydrogen bonding contacts and/or disulfide bridges to help with monomer association and tetramer assembly.[27] AQP proteins are ubiquitously found as tetramers, but each monomer constitutes a functionally independent pore.[28] We used SoPIP2;1 monomers to complete this study due to the computational efficiency and literature-based justification (see Methods – System Assembly). Given the dependence on loop A dynamics, it is likely that each of these systems would better stabilize the SoPIP2;1 crystal structures when modeled as a tetramer. However, the difference in tIC components reinforces how some bilayers can better stabilize SoPIP2;1 than others.”

[26] - Roche, J. V. & Törnroth-Horsefield, S. Aquaporin protein-protein interactions. *Int. J. Mol. Sci.* **18**, 2255 (2017).

[27] - Gössweiner-Mohr, N., Siligan, C., Pluhackova, K., Umlandt, L., Koefler, S., Trajkovska, N. & Horner, A. The hidden intricacies of aquaporins: Remarkable details in a common structural scaffold. *Small* **18**, 2202056 (2022).

[28] - Ozu, M., Alvear-Arias, J. J., Fernandez, M., Caviglia, A., Pena-Pichicoi, A., Carrillo, C., Carmona, E., Otero-Gonzalez, A., Garate, J. A., Amodeo, G. & Gonzalez, C. Aquaporin gating: A new twist to unravel permeation through water channels. *Int. J. Mol. Sci.* **23**, 12317 (2022).

Within our Methods “System Assembly” subsection, we also cite some literature discussing the functionality of monomeric aquaporin:

“The computational efficiency associated with the use of monomer SoPIP2;1 is further justified by experimental findings and related literature reviews. AQP monomers are known to constitute functionally independent pores.[28] Each AQP monomer can be functionally reconstituted *in vitro*,[74,75] and even functional AQP monomers have been resolved by solid state NMR spectroscopy (PDB IDs: 81HD and 6POJ).[76,77]”

[74] - Borgnia, M., Nielsen, S., Engel, A., Agre, P. Cellular and molecular biology of the aquaporin water channels. *Annu. Rev. Biochem.* **68**, 425–458 (1999).

[75] - Schmidt, V. & Sturgis, J. N. Making monomeric Aquaporin Z by disrupting the hydrophobic tetramer interface. *ACS Omega* **2**, 3017–3027 (2017).

[76] - Tan, H., Zhao, Y., Zhao, W., Xie, H., Chen, Y., Tong, Q. & Yang, J. Dynamics properties of membrane proteins in native cell membranes revealed by solid-state NMR spectroscopy. *Biochim. Biophys. Acta. Biomembr.* **1864**, 183791 (2022).

[77] - Dingwell, D. A., Brown, L. S. & Ladizhansky, V. Structure of the functionally important extracellular loop C of human Aquaporin 1 obtained by solid-state NMR under nearly physiological conditions. *J. Phys. Chem. B* **123**, 7700–7710 (2019).

- **Are the simulations long enough to allow the membrane to equilibrate? Esp for the complex membrane, it would be good to make sure equilibration has occurred, as 1**

microsecond might not be long enough for this. This might partially explain things like the hydrophobic mismatch.

We performed an area per lipid (APL) calculation for each of the 1 μ s production runs to test for convergence. We found that the APL converged to within a 1 \AA range after \sim 200 ns of simulation. During our analyses which used continuous trajectories to represent metastable states – such as the hydrophobic mismatch analysis – all such trajectories were adaptively acquired after the initial production run. We did not use any of the 1 μ s data runs for analyses that involved continuous trajectories.

We provide our APL calculation (Supplementary Figure 17) below. It is customary for APL calculations to be reported as either a rolling or cumulative average:

Supplementary Figure 17. Area per lipid (APL) cumulative average from each initial 1μs production run.

- As a limitation/future direction, it's worth discussing the role of asymmetry in model membranes, as discussed in a recent preprint, which suggests a strong asymmetry of PLS and sterols in the membrane (10.1101/2023.07.30.551157).

Thank you for bringing this preprint to our attention, as it is pertinent to the surprising results surrounding our complex bilayer simulations at the time of peer review.

As a summary, Doktorova and the Leventals designed a *tour de force* to defy textbook assumptions about membrane leaflet organization. The authors go to great lengths to prove that not only are bilayer compositions asymmetric, but that they re-equilibrate to preserve a delicate balance of asymmetry. Creative simulation setups and analyses were expertly blended with experiments, including newly developed experimental assays to help measure phospholipid and sterol distributions under membrane reorganization events. We have discussed their main points in our manuscript as follows.

First, our complex bilayer satisfies the criteria of a realistic bilayer with its 48% sterol composition. Therefore, our results from the complex bilayer simulations are physically valid.

In our initial draft, we recommended that “one should reduce the relative sterols composition in a complex bilayer below the threshold for crystallization or use homogenous bilayers containing lipids with polyunsaturated tails (PL or LL) and charged headgroup (PG) or zwitterionic, non-bulky headgroup (PE)” to aim for simulation of functional SoPIP2;1. Knowing that a realistic bilayer must have ≥ 40 mol% sterol in order to maintain asymmetric bilayer integrity, our previous suggestion of reducing total sterol concentration is ill-advised. While the tight packing caused by the sterol crystallization was also observed in Doktorova et al.’s work as a hallmark of cellular bilayers, it proved to inhibit SoPIP2;1 water transport function when computationally modeled.

Another suggestion made in the original submission was that “[o]ne should also consider employing multiple replicates with varied membrane packings to overcome the possibility of skewed samplings.” The idea behind using alternately constructed initial bilayers would be to disrupt some of the tight packing that may prove inhibitory towards model/simulation results. In comparison to Doktorova et al., this would be the computational equivalent to bilayer regulation by scramblase activity. To this end, the leaflet-specific stoichiometry would be preserved but greater protein dynamics could be observed.

To contextualize this work in our final manuscript, we make the following changes:

1. For the final paragraph from the Membrane Properties results section, we write:
“The complex bilayer, with a high sterol composition of 48%, completely impedes water transport in all SoPIP2;1 macrostates. A sterol composition ≥ 40 mol% is required to maintain bilayer integrity in realistic cell membranes.[58] Therefore, for SoPIP2;1, to achieve a stable, expected behavior of transport and conformational dynamics, one should disrupt sterol crystallization by diversifying the initial bilayer configuration or use homogeneous bilayers containing lipids with polyunsaturated tails (PL or LL) and charged headgroup (PG) or zwitterionic, non-bulky headgroup (PE).”
2. We have also contextualized these results into our Conclusion:
“ ... At a minimum, a composition that represents some average of the ensemble properties of a more complex or realistic bilayer should be used.

One should also consider employing replicates with varied membrane packings to overcome the possibility of skewed samplings. Even when starting with different initial membrane packings for the open and closed states, non-productive SoPIP2;1 states were still observed in this study. Asymmetric sterol distribution is necessary for biological bilayer integrity, and realistic bilayers possess tightly packed aggregate of sterols.[58] The objective of the MD practitioner now becomes equipping membrane protein simulations with a realistic, or appropriate, bilayer choice that enables functionally-relevant discovery while respecting real-life biological constraints. Our conclusion is especially important for researchers running short membrane protein simulations, as initial membrane packings are not likely to significantly deviate along the nanosecond timescale. This means that our lipid-induced SoPIP2;1 inhibition does indeed happen in cells, and that efforts must be made to also observe functional dynamics during simulation. Membrane mixing offers a solution and computational analogy to lipid imbalance regulation done by flippases and scramblases.[63] Membrane mixing will also redistribute chemically identical, but biophysically distinct, lipid molecules, which may alleviate rigidity within annular shell arrangements (Figure 8). Tools for altering membrane configuration have since become widely available to further diversify membrane packings.[13,64,65]”

[58] – Doktorova, M., Symons, J. L., Zhang, X., Wang, H.-Y., Schlegel, J., Lorent, J. H., Heberle, F. A., Sezgin, E., Lyman, E., Levental, K. R. & Levental, I. Cell membranes sustain phospholipid imbalance via cholesterol asymmetry. *bioRxiv* (2023), Preprint at <https://www.biorxiv.org/content/10.1101/2023.07.30.551157v1>.

[63] – Sakuragi, T. & Nagata, S. Regulation of phospholipid distribution in the lipid bilayer by flippases and scramblases. *Nat. Rev. Mol. Cell. Biol.* **24**, 576–596 (2023).

[13] – Lee, J., Patel, D. S., Stahle, J., Park, S.-J., Kern, N. R., Kim, S., Lee, J., Cheng, X., Valvano, M. A., Holst, O., Knirel, Y. A., Qi, Y., Jo, S., Klauda, J. B., Widmalm, G. & Im, W. CHARMM-GUI Membrane Builder for complex biological membrane simulations with glycolipids and lipoglycans. *J. Chem. Theory Comput.* **15**, 775–786 (2019).

[64] – Schott-Verdugo, S. & Gohlke, H. PACKMOL-Memgen: A simple-to-use, generalized workflow for membrane-protein–lipid-bilayer system building. *J. Chem. Inf. Model.* **59**, 2522–2528 (2019).

[65] – Licari, G., Dehghani-Ghahnaviyeh, S. & Tajkhorshid, E. Membrane Mixer: A toolkit for efficient shuffling of lipids in heterogeneous biological membranes. *J. Chem. Inf. Model.* **62**, 986–996 (2022).

- **In Fig 3A – the “closed” transporter transports more water than “open” for POPE, POPC, PLPC, LLPG. Does this suggest that these lipids are giving qualitatively incorrect data?**

Our simulation results indicate two functional constraints determine the rate of SoPIP2;1 water transport. One of the constraints is the loop D conformation, which has been the leading hypothesis speculated to govern SoPIP2;1 and PIP aquaporin regulation (Törnroth-Horsefield *et al.*, 2006). The other constraint is the potential hydrophobic rearrangement involving Ile100 (HOLE analyses; Figure 4 in the original submission, now Figure 6). It is speculated that AQP proteins block water transport by occluding the channel's intracellular cavity via side chain rearrangements, suggesting that this result has confirming support from the AQP community (Ozu *et al.*, 2022). Unfortunately, AQP water fluxes cannot be experimentally measured directly. It thus is difficult to provide a “hard” or “correct” delineation between what the contribution of loop D versus channel side chain dynamics in maintaining SoPIP2;1 water transport ability. To this exact point, “open” states for POPE, POPC, and PLPC have an open loop D yet water transport is inhibited.

We have used different lipid bilayers to show that we observe different dynamics based on the difference in lipid choice. These discrepancies in the water transport analyses could be because of the actual effect introduced by the lipids. Alternatively, this could be a methodological issue stemming from system setup. We began our simulations using available crystal structures (PDB IDs: 2b5f and 1z98), which were resolved in different lipid bilayer/detergent conditions. As such, we are limited to the starting configurations – loop D conformation and sidechain rotamer arrangements – of these crystal structures. Sadly, the input structure is a fundamental limitation to molecular dynamics simulations.

After looking through our data during manuscript preparation, channel residue rearrangement within different lipid bilayer embeddings ended up being a significant result. Given simulation times routinely employed in literature as well as in this study, what we do have are *accessible* open states. As we state in the manuscript, there is potential for the adaptive sampling protocol to have selected “conformationally trapped” states that were structurally desirable based off measurable loop D conformation. Likewise, there is the potential that more sampling could lead to more *functional* open states, as we have observed trajectories where the channel residue rearrangements can be reversed. Different bilayers can be appropriate for simulation, as shown from our hydrophobic mismatch analyses (original submission, Figure 5; now Figure 7).

We offer intriguing instances where the choice of modelled lipid bilayer can inhibit water transport in open states versus still allow water transport in closed states. As we stated in response to your suggestion about the work of Doktorova *et al.*:

“The objective of the MD practitioner now becomes equipping membrane protein simulations with a realistic, or appropriate, bilayer choice that enables functionally-relevant discovery while respecting real-life biological constraints. ... This means that our lipid-induced SoPIP2;1 inhibition does indeed happen in cells, and that efforts must be made to also observe functional dynamics during simulation.”

While lipid-based inhibition will happen in cells, modeling and simulation efforts need to be adjusted so that inhibited protein states are not the only ones to be sampled computationally.

Minor:

- **Fig 1B – the lipids are a bit too small to see**
- **Same for the labels in 1C (maybe use a shared axis?). Can the name of the lipid be added in large print on the graph?**
- **On 1c, the red and blue dots could be clearer – maybe bigger and not a colour used in the heatmap? (i.e., black or white perhaps)**
- **2c and 3b labels are too small**

We have addressed all these issues by splitting Figure 1 into three separate figures. We increased the font sizes accordingly. All remaining Figure numbers have been adjusted accordingly. The lipids were made bigger. The free energy landscape font was increased. The dots on the free energy landscapes were made larger and the black outline was thickened. The labels for original submission Figures 2c and 3b have been increased in size.

Figure 1: System compositions of SoPIP2;1 molecular dynamics simulations. (a) Crystal structures of the open (PDB ID: 2B5F, red) and closed (PDB ID: 1Z98, blue) states of spinach aquaporin in the ribbon representation. Key differences between the structures are indicated, including the “plug” residue Leu197 (shown in the stick representation) and loop D. **(b)** Chemical structures of the lipids used and their composition in the complex lipid bilayer (if present). Enclosed in the box are the lipids used in the homogeneous bilayer systems, covering all three headgroups and varying levels of acyl chain unsaturation.

Figure 2: Free energy of the SoPIP2;1 opening/closing transition from simulations. MSM-weighted (stationary distribution applied) energy landscapes of SoPIP2;1 conformational changes in lipid bilayer systems (nine homogeneous and one complex membrane) projected onto the first two components of the time-lagged independent component analysis (tICA). The clusters most similar to the open and closed crystal structures are located on the landscapes as a red and blue dot, respectively. The distance feature most correlated to each component is indicated on the axes labels and located on the crystal structure to the right of each landscape. The residues involved in the first and second tICs are shown in the blue and pink representations, respectively. Loop D is highlighted in orange as a reference.

Figure 3: Mean first passage time (MFPT) of SoPIP2;1 opening/closing transitions. MFPT of the transition between the crystal structures in the landscapes of Figure 2. Error bars are reported as the standard error of the mean of 200 bootstrapped samples.

Figure 4: Characterization of pore plug Leu197 in each of the respective SoPIP2:bilayer macrostates. (a) Atoms used in the dihedral calculation shown on the crystal structures of the open (PDB ID: 2B5F, pink) and closed (PDB ID: 1Z98, blue) states. Residues are shown in the stick representation, and atoms involved in the calculations are shown in the ball representation. (b) Simplified schematics of the dihedral angle between the C_γ of Leu197 and C_β of Ala182. (c) Violin plots of the dihedral angle in each macrostate of each lipid bilayer system.

Figure 5: Water transport activity of SoPIP2;1. (a) Average number of waters transporter per 100-ns trajectory for the open-like (yellow) and closed-like (purple) SoPIP2;1 macrostates. Errors are calculated as standard deviations among the closed-like or open-like trajectories. (b) Time evolution of the average number of waters occupying the protein pore of each macrostate in each bilayer system.

Reviewer #2 (Remarks to the Author)

Review of NCOMSS-23-33126

This is an interest manuscript using molecular dynamics (MD) simulations to explore the possible functional regulation of a plant Aquaporin (SoPIP2;1, henceforth SoPIP for short) by membrane lipids. As noted in the abstract this Aqp has no reported high-affinity lipid interactions, although this may reflect e.g. the age of the SoPIP X-ray structures – lipid like density has not been observed in a number of membrane protein structures determined recently by cryo-EM including e.g. AQP2. Furthermore, older simulations have been used to explore Aqp/phospholipid interactions (see e.g. doi: 10.1016/j.str.2013.03.005) and have been shown to reveal lipid binding sites seen in a handful of early high resolution structures of Aqp0 and Aqp4. Indeed, it is puzzling that this earlier literature is not reviewed in the introduction to the current ms.

We apologize that this earlier literature was not reviewed in the Introduction of our original submission. While our lab traditionally works on membrane transporter proteins and GPCRs, we have a general interest in selectivity and functional regulation of membrane proteins. As such, while using an aquaporin protein like SoPIP2;1 was essential for the aims of this study, we admit that we are new to the aquaporin literature. When surveying literature during this study's conception, we mainly wished to assess what type of membrane protein would work best to computationally evaluate the functional regulation associated with lipid bilayer embedding. We sincerely apologize for this oversight. We have now worked to incorporate some of the earlier aquaporin simulation literature which is relevant to this manuscript.

Such additional and related sources include:

[35] - Hall, J. E., Freitas, J. A. & Tobias, D. J. Experimental and simulation studies of Aquaporin 0 water permeability and regulation. *Chem. Rev.* **119**, 6015–6039 (2019).

[36] - Kite, R. K., Li, Z. & Walz, T. Principles of membrane protein interactions with annular lipids deduced from aquaporin-0 2D crystals. *EMBO J.* **29**, 1652–1658 (2010).

[37] - Hall, J. E., Freitas, J. A. & Tobias, D. J. Lipid membranes with a majority of cholesterol: Applications to the ocular lens and Aquaporin 0. *J. Phys Chem. B.* **115**, 6455–6464 (2011).

[38] - Aponte-Santamaria, C.; Briones, R.; Schenk, A. D.; Walz, T.; de Groot, B. L. Molecular driving forces defining lipid positions around aquaporin-0. *Proc. Natl. Acad. Sci. USA* **25**, 9887–9892 (2012).

[39] - Stansfeld, P. J., Jefferys, E. E. & Sansom, M. S. P. Multiscale simulations reveal conserved patterns of lipid interactions with aquaporins. *Structure* **21**, 810–819 (2013).

[40] - Newport, T. D., Sansom, M. S. P. & Stansfeld, P. J. The MemProtMD database: A resource for membrane-embedded protein structures and their lipid interactions. *Nucleic Acids Res.* **47**, D390–D397 (2019).

[49] - Briones, R., Aponte-Santamaria, C. & de Groot, B. L. Localization and ordering of lipids around Aquaporin-0: Protein and lipid mobility effects. *Front. Physiol.* **8**, 124 (2017).

[111] - Song, W., Corey, R. A., Ansell, T. B., Cassidy, C. K., Horrell, M. R., Duncan, A. L., Stansfeld, R. J. & Sansom, M. S. P. PyLipID: A Python package for analysis of protein-lipid interactions from molecular dynamics simulations. *J. Chem. Theory Comput.* **18**, 1188-1201 (2022).

Having said that, SoPIP is an interesting candidate for further investigation of Aqp/lipid interactions, given that it shows regulation (gating) of water flow by the conformation of a cytosolic loops which is not seen in other Aqps. The authors therefore focus on possible lipid effects on such regulation.

We thank Reviewer 2 for seeing purpose in our work.

All atom simulations were performed for SoPIP in nine different single lipid bilayers and one more complex mixed lipid bilayer. The simulations were in the presence of 0.2 M Ca ions. To what extent are we confident that the CHARMM36 forcefield reproduces lipid/Ca ion interactions accurately? I am aware of several simulation studies attempting to improve Ca ion interactions with lipids.

In 2016, the calcium parameters in the CHARMM36 force field were revisited by CHARMM developers Mohsen Pourmoussa, Richard Venable and Richard Pastor. Specifically, the Lennard-Jones parameters in the CHARMM36 force field “for pairwise interactions of calcium ions with chloride and with negatively charged groups of phospholipids” were changed. According to their abstract describing the implemented force field change:

“This [was] achieved by simulating osmosis and electrophoresis phenomena for calcium salts and matching simulation results and experimental data. The parameters [were] then validated by simulating lipid bilayers in mixtures of calcium chloride and sodium chloride solutions and comparing the bilayer properties with experimental data.” (Pourmoussa, Venable & Pastor, 2016)

A recent Richard Venable and Richard W. Pastor PNAS paper from 2022 used the CHARMM36 force field to perform simulations of lipid-bilayer containing systems with divalent calcium ions. In their Materials and Methods section, they reiterate how the study used “existing pair-specific NBFIX (nonbonded fix) LJ parameters of Ca^{2+} with Cl^- and with the oxygen atom of P=) bond on the phosphate groups P1, P4, and P5.

Like their study, we also used CHARMM-GUI to build our systems during the Spring of 2022. As such, we have made use of similar “existing pair-specific NBFIX LJ parameters”. Therefore, we have confidence that our use of the CHARMM36 force field reproduces lipid/ Ca^{2+} ion interactions accurately.

We have edited our methods section to convey the correct use of CHARMM36 force field parameters. We have also included all such above studies as citations:

Revision:

“Force field parameters were CHARMM36⁸³ with existing pair-specific NBFIX (nonbonded fix) Lennard-Jones parameters for Ca²⁺ to Cl⁻ ion pairing, as well as Ca²⁺ and phosphate group-oxygen atom pairings.[84-86]”

[84] – Pourmousa, M., Venable, R. M. & Pastor, R. W. Calcium parameters in CHARMM force field revisited. *Biophys. J.* **110**, 327a–328a (2016).

[85] – Han, K., Venable, R. M., Bryant, A.-M., Legacy, C. J., Shen, R., Li, H., Roux, B., Gericke, A. & Pastor, R. W. Graph–theoretic analysis of monomethyl phosphate clustering in ionic solutions. *J. Phys. Chem. B* **122**, 1484–1494 (2018).

[86] – Han, K., Kim, S. H., Venable, R. M. & Pastor, R. W. Design principles of PI(4,5)P₂ clustering under protein-free conditions: Specific cation effects and calcium-potassium synergy. *Proc. Natl. Acad. Sci. USA* **119**, e2202647119 (2022).

Multi-microsecond simulations, adaptive state sampling, and Markov state models were employed in the simulations. This is a careful, state of the art approach. Analysis focused on the functional dynamics of the protein, especially the pore plug, and effects on water transport.

We thank Reviewer 2 for appreciating our work, its construction and scope, and the methodology needed to deliver its main ideas to you through this peer review process. As a general comment, we also thank you for engaging us with comments based upon scientific discussion.

I have two main questions which need to be addressed.

- 1. Are there any experimental data for lipid effects on SoPIP function which correlated with the simulation results?**

Unfortunately, we could not find any literature references specific to SoPIP function that experimentally demonstrate lipid effects on the protein. We do, however, already mention a few studies that correlate with our results. For example, during discussion of simulated water transport function, we had written:

“Overall, the computed number of water molecules transported is within three orders of magnitude from the experimental literature value of 10⁴ waters/100 ns for SoPIP₂;1 in the *E.coli* liposome[31] or 100-200 waters/100 ns for aquaporin in general.[32,33]”

These reported rates of simulated water transport are reasonable, especially considering that our simulations involve monomer SoPIP₂;1, whereas these experimental results implicate tetramer function. Similarly, our original submission references aquaporin literature about the affects of

sterols on aquaporin function, when we talk about functional depletion of SoPIP₂;1 water transport when simulated using a realistic plant plasma membrane bilayer:

“Additionally, experimental studies have shown that the addition of sterols reduces water transport in AQP0[56] and AQP4.[57]”

[31] – Kirscht, A., Survery, S., Kjellbom, P. & Johanson, U. Increased permeability of the aquaporin SoPIP₂;1 by mercury and mutations in loop A. *Front. Plant Sci.* **7** (2016).

[32] – Smolin, N., Li, B., Beck, D. A. C. & Daggett, V. Side-chain dynamics are critical for water permeation through aquaporin-1. *Biophys. J.* **95**, 1089–1098 (2008).

[33] – Binesh, A. R. & Kamali, R. Molecular dynamics insights into human aquaporin 2 water channel. *Biophys. Chem.* **207**, 107–113 (2015).

[56] – Tong, J., Canty, J. T., Briggs, M. M. & McIntosh, T. J. The water permeability of lens aquaporin-0 depends on its lipid bilayer environment. *Exp. Eye Res.* **113**, 32–40 (2013).

[57] – Tong, J.; Briggs, M. M.; McIntosh, T. J. Water permeability of aquaporin-4 channel depends on bilayer composition, thickness, and elasticity. *Biophys. J.* **103**, 1899–1908 (2012).

2. Was any analysis of SoPIP/lipid interactions performed? Are any lipid binding sites comparable to those for other Aqps (see comment above) or for other membrane proteins observed?

Following Reviewer 2’s suggestion, we proceeded with a systematic comparison of SoPIP₂;1 lipid binding sites against what has been recorded for other aquaporins. Some of the more foundational literature that we had uncovered during this peer review hinted that lipid electron densities in resolved AQP structures were representative of the ensemble average of aquaporin-lipid interactions. Furthermore, many MD simulation studies were able to spontaneously replicate annular shell arrangements seen in resolved AQP structures. As such, we relied on the MemProtMD database for comparing SoPIP₂;1 lipid binding to other aquaporin lipid binding events. We describe our protocol below in our new Methods subsection “Lipid Binding Site Analysis”:

“Generally, AQP proteins are known to have conserved lipid binding sites. To compare how lipid binding occurred throughout SoPIP₂;1 loop D opening/closing transitions, PyLipID software was used to calculate lipid-protein residue interactions.[111] The default lipid contact distance cutoff parameters of 4 to 8 Å were used as suggested by PyLipID tutorials. Calculated results were averaged with residence times reported in nanoseconds and presented as heatmaps. For SoPIP₂:bilayer embeddings where more than one metastable intermediate state existed, the intermediate state residence times were averaged to determine the resulting heatmap. For reporting state-specific loop D interaction results, radial bar graphs were used to generate “fingerprints” with the resulting PyLipID residence time data. Because the SoPIP₂:complex

simulations involve different lipid species, the maximum SoPIP2:lipid interaction residence time was reported for each residue.

After the binding site calculations were performed on continuous trajectories representing each of the metastable states, the results were compared to generally observed trends in previously reported AQP lipid binding site analyses. Specifically, we compared against results from multiscale simulations performed by Stansfeld, Jefferys, and Sansom,[39] as well as the MemProtMD database.[40] We first retrieved each AQP tetrameric structure from the MemProtMD database except for PDB IDs 2B5F and 1Z98, totaling 52 unique AQP PDB IDs. Under visual inspection, each of the 52 tetramers was divided into monomeric units. Because our SoPIP2;1 simulations were performed in the monomeric form, we focused our analysis by only focusing on potential lipid binding sites that would be seen under the SoPIP2;1 tetramer assembly. Using the MemProtMD 2B5F and 1Z98 models, we identified four potential lipid binding sites based off lipid-orientation. These four lipid-exposed stretches on SoPIP2;1 included (1) Asp28 to Tyr53, (2) Phe86 to Gln147, (3) Gly158 to Phe204, and (4) Pro223 to Val263.

With the putative SoPIP2;1 tetramer binding sites identified, each monomeric protomer separated from the 52 MemProtMD tetramers was structurally aligned to a SoPIP2;1 monomer. After structural alignment, multiple sequence alignments were generated using only structurally aligned MemProtMD monomer PDB files as input for Promals3D.112 Resulting multiple sequence alignments were analyzed to determine where SoPIP2;1-homologous lipid-exposed stretches existed on each MemProtMD monomer. These sequence mappings were visually confirmed for each of the 208 monomer structures. Once the sequence mappings were structurally confirmed, the maximum of the MemProtMD lipid head and tail contact probabilities were retained for each monomer residue. The resulting lipid contact probability for each structurally equivalent residue was averaged, yielding a singular per-residue value for each of the 52 MemProtMD structures.

To compare our results against the MemProtMD structures, SoPIP2;1 per-residue lipid PyLipID residence times were averaged across all representative open, intermediate, and closed state trajectories and then reported as a contact probability.”

We then described these results in our new Results subsection “Lipid Binding Interactions”:

“Whether intracellular rearrangements seen in nonfunctional SoPIP2;1 states were caused by direct protein-lipid interactions was examined. With the pioneering advances in membrane protein simulation of the 1990s and the beginning of AQP structure determination in the early 2000s, there is a rich computational history on AQP-lipid interactions.[35] Foundational simulation studies examining the AQP-lipid interface have found protein-lipid interactions to be consistent with crystal and electron densities, showing conservation of specific protein-lipid interaction sites without high-specificity binding.[36-39] Lateral exchange of annular shell lipids with the bulk has therefore been suggested as a regulatory mechanism for AQP function.[39] Conserved channel residue rearrangement have also been posited as a regulatory

means to affect AQP water transport.[28,39] Perhaps lipid binding itself could be the fundamental driver of either proposed regulatory mechanism.

Comparing lipid binding interactions from our 100-ns metastable state trajectories, SoPIP2;1 was found to have an identical lipid binding interface compared to 52 previously simulated AQP proteins (Supplementary Figures 21-23).[40] Likewise, high-specificity lipid binding sites are also presumed to be absent in SoPIP2;1. Average lipid residence times between SoPIP2;1 open, closed, and intermediate states across the lipid bilayers are nearly identical as well (Supplementary Figures 24-26). Out of the residues involved in the inhibitory channel rearrangements, only Leu197 and Ile 202 have consistent interactions with lipids. Meanwhile, Leu175 has moderate lipid interactions except for POPC, POPG and LLPC simulations.

Given that Leu197 demonstrated strong interactions with each lipid bilayer, we generated state-specific loop D-lipid interaction fingerprints (Supplementary Figure 27). Nearly all bilayers maintain strong interactions with loop D residues Leu197 and Ala198, showing average residence times greater than 70 ns. Loop D maintains the most lipid interactions when surrounded by POPE, followed by PLPE. Otherwise, most bilayer compositions only strongly interact with ~5 out of the 16 loop D residues. While SoPIP2;1 is in the closed state, loop D lipid interactions are minimal, as the loop is occluding the pore. Overall, our resulting fingerprints demonstrate diverse binding signatures arise when using different bilayers.”

The Supplementary Figures relating to this section are presented below:

Supplementary Figure 21. Comparison of average SoPIP2;1 lipid contact probability along the protein-lipid interface against MemProtMD simulated aquaporins (Part 1). The colorbar reports lipid contact probability as a percentage. Black squares in the heatmap indicate aquaporin residues which did not structurally align to any portion of SoPIP2;1 with a one-to-one mapping.

Supplementary Figure 22. Comparison of average SoPIP2;1 lipid contact probability along the protein-lipid interface against MemProtMD simulated aquaporins (Part 2). The colorbar reports lipid contact probability as a percentage. Black squares in the heatmap indicate aquaporin residues which did not structurally align to any portion of SoPIP2;1 with a one-to-one mapping.

Supplementary Figure 23. Comparison of average SoPIP2;1 lipid contact probability along the protein-lipid interface against MemProtMD simulated aquaporins (Part 3). The colorbar reports lipid contact probability as a percentage. Black squares in the heatmap indicate aquaporin residues which did not structurally align to any portion of SoPIP2;1 with a one-to-one mapping.

PyLipID residence time SoPIP2^{open}

Supplementary Figure 24. PyLipID average residence time for SoPIP2:lipid interactions along the open state. The colorbar reports lipid residence time in nanoseconds based off calculations performed on continuous trajectories representing metastable states.

Supplementary Figure 25. PyLipID average residence time for SoPIP2:lipid interactions along intermediate states. The colorbar reports lipid residence time in nanoseconds based off calculations performed on continuous trajectories representing metastable states.

PyLipID residence time SoPIP2^{closed}

Supplementary Figure 26. PyLipID average residence time for SoPIP2:lipid interactions along the closed state. The colorbar reports lipid residence time in nanoseconds based off calculations performed on continuous trajectories representing metastable states.

Supplementary Figure 27. Radial fingerprints for loop D lipid binding interactions. The angular axis depicts bar spokes representing each residue in loop D. The radial axis describes the PyLipID residence time for which lipids in each bilayer are bound to the selected residue, expressed in nanoseconds. Data representing each metastable state identity are colored accordingly.

Taken together these are important as the lipid analysis in the manuscript concentrate on bilayer biophysical properties e.g. possible hydrophobic mismatch. The authors conclude (page 28) that “the lipids used in this study can adapt to SoPIP2;1”. I am perhaps not surprised that the local thickness of a bilayer adapts to the structure of a relatively rigid membrane protein such as an Aqp, rather than vice versa (as might be expected for a protein which could adopt different conformation within a bilayer, e.g. an ion channel or a transporter). Given the regulation of SoPIP involves a cytosolic loop I would have expected a more detailed examination of specific loop/lipid interactions.

The authors state that “Overall, the transition path from closed to open varies drastically for the same protein embedded in different membrane bilayers”. Given this I would have expected at least some search of specific lipid interactions. Even if these were not observed, they could then be ruled out as a mechanism, thus refocusing attention on bilayer biophysical properties such as local thickness.

From our new aquaporin-lipid interactions, we actually found that the areas to which lipid binding varies most considerably between all the SoPIP:lipid embeddings is along loop D. However, given that the overall lipid binding profiles are virtually identical between all bilayer embeddings, we still hold that the ensemble biophysical properties of the bilayer still hold more importance than direct contacts.

This same idea was proposed in earlier AQP simulation literature, where the work “proposed bilayer thickness and ordering within the annular shell as a mechanism for regulating water transport.⁴⁹” This notion becomes especially apparent given our results, as nearly no residues involved in the rearrangement leading to hydrophobic water blockage have direct lipid contacts. Additionally, that “[o]ur calculations of reduced water transport agree with our loop D fingerprint analysis, as the stiffened POPE bilayer sustains many long-lasting loop D interactions (Supplementary Figure 27).”

We thank Reviewer 2 for their suggestions. By comparing our SoPIP2;1 results to all other AQP proteins, we are able to extend our findings to suggest that the adaptation of bilayer properties could act as a more universal form of membrane protein regulation. From our literature search needed for addressing all questions for this peer review, we are happy to see our work satisfy scientific needs within the AQP community and the membrane protein community at large.

Rebuttal Letter References

- Ozu, M. et al. Aquaporin gating: A new twist to unravel permeation through water channels. *Int. J. Mol. Sci.* **23**, 12317 (2022).
- Tan, H. et al. Dynamics properties of membrane proteins in native cell membranes revealed by solid-state NMR spectroscopy. *Biochim. Biophys. Acta. Biomembr.* **1864**, 183791 (2022).
- Dingwell, D. A., Brown, L. S. & Ladizhansky, V. Structure of the functionally important extracellular loop c of human aquaporin 1 obtained by solid-state NMR under nearly physiological conditions. *J. Phys. Chem. B* **123**, 7700–7710 (2019).
- Borgnia, M., Nielsen, S., Engel, A. & Agre, P. Cellular and molecular biology of the aquaporin water channels. *Annu. Rev. Biochem.* **68**, 425–458 (1999).
- Schmidt, V. & Sturgis, J. N. Making monomeric aquaporin z by disrupting the hydrophobic tetramer interface. *ACS Omega* **2**, 3017–3027 (2017).
- Gössweiner-Mohr, N. et al. The hidden intricacies of aquaporins: Remarkable details in a common structural scaffold. *Small* **18**, 2202056 (2022).
- Roche, J. V. & Törnroth-Horsefield, S. Aquaporin protein-protein interactions. *Int. J. Mol. Sci.* **18**, 2255 (2017).
- Doktorova, M. et al. Cell membranes sustain phospholipid imbalance via cholesterol asymmetry. Preprint at <https://www.biorxiv.org/content/10.1101/2023.07.30.551157v1> (2023).
- Törnroth-Horsefield, S. et al. Structural mechanism of plant aquaporin gating. *Nature* **439**, 688–694 (2006).
- Pourmousa, M., Venable, R. M. & Pastor, R. W. Calcium parameters in CHARMM force field revisited. *Biophys. J.* **110**, 327a–328a (2016).
- Han, K., Kim, S. H., Venable, R. M. & Pastor, R. W. Design principles of PI(4,5)p₂ clustering under protein-free conditions: Specific cation effects and calcium-potassium synergy. *Proc. Natl. Acad. Sci. USA* **119**, e2202647119 (2022).

Reviewer #1 (Remarks to the Author):

The authors have thoroughly and satisfactorily answered my queries, so now I fully support publication.

Reviewer #2 (Remarks to the Author):

The authors have worked hard to address the concerns raised in my earlier review. All of my main points have been addressed in detail.

My only remaining reservation concerns the absence of any experimental data for lipid effects on SoPIP2 function.

As the authors state: "we could not find any literature references specific to SoPIP function that experimentally demonstrate lipid effects on the protein".

However, this does not necessarily preclude publication of a detailed and insightful simulation study of predicted lipid effects on SoPIP2.

REVIEWER COMMENTS – FINAL ROUND OF REVISION

Reviewer #1 (Remarks to the Author)

The authors have thoroughly and satisfactorily answered my queries, so now I fully support publication.

Thank you for your support of our publication. It is greatly appreciated. We are thankful for your review, especially as it enabled important discussion concerning our findings.

Reviewer #2 (Remarks to the Author)

The authors have worked hard to address the concerns raised in my earlier review. All of my main points have been addressed in detail.

My only remaining reservation concerns the absence of any experimental data for lipid effects on SoPIP2 function. As the authors state: “we could not find any literature references specific to SoPIP function that experimentally demonstrate lipid effects on the protein.” However, this does not necessarily preclude publication of a detailed and insightful simulation study of predicted lipid effects on SoPIP2.

We are very appreciative of Reviewer #2's persistence on this matter. It turns out our original search was naïve and focused only on how lipid effects contribute to observed water transport rates. We reformed our query on Google Search and WebOfScience using the terms “SoPIP2” and “lipid”. In this way, we were indeed able to find experimental evidence concerning lipid effects on our model protein system.

The sources we found include the following, numbered by how they appear within our manuscript's final draft:

[59] – Plasencia, I. *et al.* Structure and stability of the spinach aquaporin SoPIP2;1 in detergent micelles and lipid membranes. *PLOS ONE* **6**, e14674 (2011).

[60] – Hansen, J. S. *et al.* Interaction between sodium dodecyl sulfate and membrane reconstituted aquaporins: A comparative study of spinach SoPIP2;1 and E. coli AqpZ. *Biochim. Biophys. Acta* **1808**, 2600-2607 (2011).

[61] – Frick, A. *et al.* Mercury increases water permeability of a plant aquaporin through a non-cysteine-related mechanism. *Biochem. J.* **454**, 491-499 (2013)

[62] – Hansen, J. S., Thompson, J. R., Hélix-Nielsen, C. & Malmstadt, N. Lipid directed intrinsic membrane protein segregation. *J. Am. Chem. Soc.* **135**, 17294-17297 (2013).

Generally, this literature reports experimental evidence that introduction of sterols is indeed detrimental to SoPIP2;1 function. Bilayers stiffened by cholesterol were shown to destabilize the SoPIP2;1 topology. To compensate, SoPIP2;1 was seen to participate in hydrophobic rearrangements between alpha-helices, which is similar to the rearrangements we had seen from nonfunctional open states with hydrophobic blockages. Functional SoPIP2;1 was seen to localize in cholesterol-poor domains of the membrane when inserted into vesicular constructs. This corroborates our simulation results where we detect nonfunctional SoPIP2;1 open states within more rigid membranes, and our explanations provided by hydrophobic mismatch and lipid order parameter analyses. In total, our findings for how membrane composition affects modeled membrane protein dynamics are strengthened. Our findings can be generalized to *in vitro* experiments.

Our new literature search has pointed out to us that understanding the basis of mechanosensitivity in the regulated function of aquaporins has been a long-standing question in the aquaporin community, which our study addresses through its predictions. Lastly, our simulation results give credence to the notion that cells can supplant existing lipid compositions to regulate membrane protein function. This final concept of lipid-based regulation proves challenging to MD simulation practitioners, as model lipid bilayers used for simulation present a fixed stoichiometry. The task remains for computational researchers to strike a balance where realistic membrane compositions are used while still yielding functionally relevant state discovery.

Our specific edits to the Main Text conclusion are as follows. Preexisting text is in red while new text is given in blue:

“ ... We uncovered how the loop D conformation does not directly alter the water conductivity of the SoPIP2;1 channel or overall transport activity due to the stabilization of a hydrophobic blockage inside the non-transporting SoPIP2;1 conformation. Literature focused on membrane protein biotechnology applications that also used SoPIP2;1 as a model system support the existence of hydrophobic blockages for non-transporting cases. Circular dichroism experiments have shown SoPIP2;1 to lose alpha-helical content and partially unfold when reconstituted into vesicle or liposome bilayers stiffened by cholesterol.^{59,60} In fact, SoPIP2;1 structurally responds to these stiffer environments with hydrophobic movements within and between alpha-helices.⁶⁰ Stopped flow experiments where mercury was used to alter membrane fluidity have shown that SoPIP2;1 transport is

affected by bilayer properties.⁶¹ Mechanosensitive bias against cholesterol by SoPIP2;1 has been further validated using fluorescence experiments, where SoPIP2;1 preferably localizes in cholesterol-poor domains.⁶²

“A general belief exists within literature that certain bilayer compositions can push equilibrium towards favorable conditions for membrane proteins. While cells can rapidly replace their lipid bilayer compositions, molecular models and simulations experience fixed stoichiometry. However, *in silico* simulated lipid molecules sample biophysical properties for direct modulation of membrane protein activity. **Hindered transport of the open SoPIP2;1 in the complex bilayer ...**”

Again, we thank Reviewer #2 for pushing us to include a proper literature search focused on anything related to SoPIP2;1 and lipid interactions. The inclusion of these biophysical references gives us more confidence in our predictions and their extension to future studies informed by our work.